

# Can low-resolution CMIP6 ScenarioMIP models provide insight into future European Post-Tropical Cyclone risk?

Elliott M. Sainsbury[1], Reinhard K. H. Schiemann[2], Kevin I. Hodges[2], Alexander J. Baker[2], Len C. Shaffrey[2], Kieran T. Bhatia[3], Stella Bourdin[4]

[1]Department of Meteorology, University of Reading, Reading, Berkshire, UK

[2]National Centre for Atmospheric Science, University of Reading, Reading, Berkshire, UK

[3]Guy Carpenter, New York, United States

[4]Laboratoire des Sciences du Climat et de l'Environnement, LSCE/IPSL, CEA-CNRS-UVSQ-Université Paris-Saclay, Gif-

sur-Yvette, France

*Correspondence to*: Elliott M. Sainsbury (e.sainsbury@pgr.reading.ac.uk)

**Abstract.** Post-Tropical Cyclones (PTCs) can cause extensive damage across Europe through extreme winds and heavy precipitation. With increasing sea surface temperatures, tropical cyclones (TC) may form and travel further polewards and eastwards than observed historically. Recent work has suggested that the frequency of intense Europe-impacting PTCs may

increase substantially in the future.

Using an objective feature tracking scheme and TC identification method, we track and identify the full lifecycle of TCs in the North Atlantic in five CMIP6 climate models in the historical (1984-2014) period and in the future under the SSP5-85 scenario (2069-2099). These five models are selected based on their ability to simulate similar TC frequency similar to observed in the North Atlantic, although model deficiencies remain.

We find no robust changes in Europe-impacting PTC frequency or intensity in the future. This is because two competing factors – a significant decrease in TC frequency of 30-60%, and an increase in the proportion of TCs reaching Europe – are approximately the same size. The projected increase in the proportion of TCs reaching Europe is largely driven by an increase in the likelihood of recurvature and is consistent with projected decreases in vertical wind shear and increases in potential intensity along the US East Coast in the future. The projected increased likelihood of recurvature is also associated with a shift

in TC genesis away from the main development region, where model biases cause very few TCs to recurve. This study indicates that large uncertainties surround future Europe-impacting PTCs and provides a framework for evaluating PTCs in future generations of climate models.





# 1 Introduction

Post-Tropical Cyclones (PTCs) can bring Europe hazardous weather such as extreme precipitation, high winds, and large waves (Bieli et al., 2019; Evans et al., 2017; Jones et al., 2003). Compared to the overall European cyclone climatology, PTCs are disproportionately responsible for the most intense windstorms to impact Europe during hurricane season (Sainsbury et al., 2020), and often attain their maximum intensity a couple of days after impacting the region, enhancing their destructive potential (Baker et al., 2021; Dekker et al., 2018).

In 2017, hurricane Ophelia (Rantanen et al., 2020) became the easternmost major hurricane on record (Stewart, 2018), and in 2019 hurricane Lorenzo became the easternmost category 5 hurricane on record. Both cyclones later impacted Europe as PTCs, and Ophelia was responsible for Ireland's highest-recorded wind gust of 53ms⁻¹ (119mph). Projected increases in sea surface temperature (SST) and the range of latitudes occupied by TCs (Studholme et al., 2022), combined with the observed trend in TC lifetime maximum intensity latitude (Kossin et al., 2014) opens the possibility for more cyclones to form– and attain high

intensities- further polewards and eastwards in the basin, closer to Europe (Baker et al., 2022; Haarsma et al., 2013; Haarsma, 2021). Additionally, as TCs attain greater intensities they may become more resilient to hostile environmental conditions such as decreasing SSTs and increasing wind shear (Baker et al., 2022; Michaelis and Lackmann, 2019), increasing their likelihood of both recurving (Sainsbury et al., 2022a), and reaching Europe (Sainsbury et al., 2022b). Diabatic processes have also been shown to be important in case studies of high-impact PTCs (Rantanen et al., 2020), implying a future increased PTC risk in a

warmer atmosphere which is capable of holding a larger amount of moisture (Haarsma, 2021).

Few studies have investigated the projected changes of Europe-impacting PTCs. Using a high resolution (~25km in midlatitudes) climate model with prescribed SSTs, Haarsma et al. (2013) found large increases in hurricane-force PTC frequency over Norway, the North Sea, and the Bay of Biscay by the end of the 21st century under the IPCC Representative Concentration Pathway 4.5 (RCP4.5) scenario, but based on a small sample size. The projected increase was associated with

an increase in SSTs, extending the TC genesis region poleward and eastwards and therefore allowing more TCs to reach the baroclinic midlatitudes before dissipating. When considering all Europe-impacting PTCs in the same simulations, the minimum sea level pressure that the cyclones attained was found to be 8 hPa lower at the end of the 21st century (Baatsen et al., 2015).

Liu et al. (2017) considered North Atlantic TCs undergoing extratropical transition (ET) more generally and found an increase

in TC density in the eastern North Atlantic under the RCP 4.5 emission scenario by the end of the century in a flux-adjusted version of the FLOR model (Vecchi et al., 2014), indicating an increase in TC-related risks for Europe. A statistically significant increase in the fraction of TCs undergoing ET is also found in the future (Liu et al., 2017). This trend has also been found in several (but not all) reanalyses (Baker et al., 2021), and mixed results have been found in climate model studies (Michaelis and Lackmann, 2019; Bieli et al., 2020). Using a pseudo global warming (PGW) approach to dynamical

downscaling, Jung and Lackmann (2019) found that hurricane Irene (Avila and Cangialosi (2011)) would be considerably stronger (> 20hPa deeper) in a future climate under the RCP 8.5 scenario and would undergo extratropical transition for





considerably (18hours) longer, extending TC-like conditions further poleward than in the present climate. Further PGW case studies also show increases in precipitation during the ET phase (Liu et al., 2020) and an increase in TC strength during the ET process (Jung and Lackmann, 2021), further highlighting the potential future increase in TC-related hazards to midlatitude

regions. Finally, Baker et al. (2022) find an increase in the frequency of ET events in the North Atlantic along with a poleward shift in ET location by 2050 in HighResMIP models under the RCP 8.5 scenario. While this is the first multi-model study of projected changes in ET in an ensemble of high-resolution climate models, it does not have a European focus. Additionally, projections are limited to 2050, by which time forced changes may not have fully emerged.

Given the potential for an increased PTC risk to Europe in the future, a multi-model study with a focus on Europe is necessary

to quantify the associated model uncertainty and to assess whether lower resolution models can provide insight into future PTC changes. Many lower-resolution climate models do not simulate a realistic TC frequency (e.g., Camargo, 2013), and even high-resolution climate models are unable to capture the strongest TCs (e.g., Walsh et al., 2015; Vidale et al., 2021; Roberts et al., 2020a). In this paper we investigate the projected changes in Europe-impacting PTCs in an ensemble of five CMIP6 models which are shown to simulate a realistic North Atlantic TC frequency compared to observations. These models have a

lower horizontal resolution than previous studies (e.g., Baker et al., 2022; Haarsma et al., 2013), and thus simulated TCs are expected to be weaker. However, a multi-model study using models containing multiple ensemble members allows for a greater sample size, and greater uncertainty quantification. It is unknown whether low-resolution CMIP6 models can give insight into projected changes in TC and PTC statistics despite their deficiencies and biases. This is investigated in this study. As far as the authors are aware, a multi-model study with a focus particularly on Europe-impacting PTCs has never been undertaken.

This paper aims to answer the following questions:

- To what extent can CMIP6 models capture the characteristics of the North Atlantic TC climatology?
- How well do CMIP6 models capture the disproportionate intensity associated with Europe-impacting PTCs relative to the overall cyclone climatology?
- Are there any areas of model consensus regarding projected changes in PTC frequency over Europe?

In section 2, we describe the cyclone detection and tracking scheme, TC identification procedure, and CMIP6 models included in this study. Section 3 contains an overview of the TC climatologies in the selected CMIP6 models, the projected changes in the frequency and intensity of Europe-impacting PTCs, and further analysis to investigate the cause of the projected changes. The paper concludes with a discussion in section 4.



## 2. Methods

### 2.1. Data

For this study, we use data from the fully-coupled historical and SSP5-85 model simulations from phase 6 of the Coupled Model Intercomparison Project (CMIP6, Eyring et al. (2016)). On average, there are 6.4 North Atlantic hurricanes (wind speeds >= 33ms⁻¹) per year in observations (HURDAT2) between 1950 and 2014. Climate models tend to underestimate TC frequency, therefore models which simulate a median TC frequency > 6.4 year⁻¹ during the North Atlantic hurricane season (June 1st – Nov 30th), averaged over the entire historical run, are selected. More information on TC identification can be found in section 2.3, and additional information on CMIP6 model selection can be found in the supplementary material (Figure S1). The chosen 5 models are: CNRM-CM6-1-HR (CNRM hereafter), HadGEM3-GC31-MM (HadGEM hereafter), KIOST-ESM (KIOST hereafter), MIROC6 (MIROC hereafter) and IPSL-CM6A-LR (IPSL hereafter). The period 1984-2014 is used for the historical run, and 2069-2099 for the SSP5-85 scenario, giving an 85-year difference between the mid points of the two time periods considered in this study. More information can be found in Table 1. Key results have been reproduced using only ensemble members which are available for both the historical and SSP5-85 scenario simulations and are shown in the supplementary material (Figure S7, Table S1). The 6-hourly wind components are utilised at 850, 500 and 250hPa for calculation of the vorticity fields necessary for TC identification (more information in section 2.2). The 6-hourly mean sea level pressure and 10m wind data are also used to investigate the intensity of the cyclones. Monthly mean temperature and specific humidity profiles are utilised to calculate potential intensity (PI), and monthly mean relative humidity, wind, and SSTs to construct the genesis potential index (Emanuel and Nolan, 2004).

| Model | Reference | Ens. members (hist/ssp585) | Atmosphere model resolution | Ocean model resolution | Vertical levels (atmos.) | Vertical levels (ocean) |
|---|---|---|---|---|---|---|
| CNRM-CM6-1-HR | (Voldoire et al., 2019) | 1/1 | ~50km | ~0.25° | 91 | 75 |
| HadGEM3-GC31-MM | (Andrews et al., 2020) | 1/1 | N216 (~60km) | ~0.25° | 85 | 75 |
| KIOST-ESM | (Pak et al., 2021) | 1/1 | ~200km | ~100km | 32 | 50 |
| MIROC6 | (Tatebe et al., 2018) | 10/3 | T85 (~1.4 degrees) | ~1° | 81 | 63 |
| IPSL-CM6A-LR | (Boucher et al., 2020) | 32/1 | ~157km | ~1° | 79 | 75 |



**Table 1. Summary of the CMIP6 models used in this study, including model name (column 1), reference to model development (column 2), number of ensemble members used (column 3), atmospheric model resolution (column 4), ocean model resolution (column 5), and number of vertical layers in the atmosphere (column 6) and ocean (column 7) models.**

Using the same tracking and TC identification scheme as CMIP6 models, the European Centre for Medium Range Weather Forecasts 5th reanalysis (ERA5, Hersbach et al. (2020)) is used for verification of model TC climatologies from 1984-2014. The 6-hourly relative vorticity fields from ERA5 are used for cyclone tracking and TC identification, and 6-hourly sea level pressure and 10m winds from ERA5 to investigate cyclone intensity.

6-hourly position, 10m wind speed and sea level pressure data from the HURricane DATabase version 2 (HURDAT2, Landsea and Franklin (2013)) is used between 1984 and 2014 in section 3.1 to compare TC intensity and spatial distribution with those simulated in ERA5 and CMIP6.

## 2.2. Cyclone tracking

Cyclone detection and tracking is performed using the objective feature tracking scheme, TRACK (Hodges, 1994, 1995, 1999), configured for TCs. The tracking is performed on the spectrally filtered (T63 resolution) relative vorticity fields vertically averaged (600-850hPa) for ERA5, and at 850hPa (spectrally filtered to T42) for CMIP6 due to data availability. For more information on the tracking scheme, see Hodges et al. (2017).

The spatial distribution of TC track and genesis densities are calculated from the cyclone tracks using the spherical kernel method described in Hodges (1996). Densities are expressed as cyclones per year per unit area, where the unit area is equivalent to a spherical cap with a radius of 5 degrees (Figures 1 and 2).

## 2.3. Objective TC identification

TCs are identified from the larger sample of tracked cyclones using the methodology of Hodges et al. (2017). Only tracks which form in the North Atlantic hurricane season (June 1st – November 30th) are considered in this study. The TC identification criteria are applied to the vorticity fields at the 850, 700, 600, 500, 400, 300 and 250hPa pressure levels for ERA5, but only to the 850, 500 and 250hPa levels for CMIP6 models due to data availability. The SSTs are expected to increase as a result of climate change, and so the poleward extent of TC genesis may move polewards, potentially beyond 30N (which is the latitude constraint placed on genesis of TCs in Hodges et al. (2017)). To ensure that our TC identification method is suitable, we first check that there is no robust projected increase in TC genesis poleward of 30N. This is achieved by modifying the TC identification method by relaxing the latitude constraint for genesis to 45N, then re-identifying TCs. We then investigate the change in genesis density between the 2069-2099 and 1984-2014 periods. No robust trend in TC genesis poleward of 30N is found (Fig. S2), and so the original TC identification method is retained.



The TC identification method used here has been used in numerous studies based on reanalyses (Hodges et al., 2017; Baker et
al., 2021) and climate models (e.g., Baker et al., 2022; Roberts et al., 2015; Vidale et al., 2021). It has been shown that PTC
impacts over Europe in the present climate are similar whether this objective TC identification method, or objective track
matching with observational data, are used (Sainsbury et al., 2020).

**2.4. Recurvature, Europe definitions and regional domains**

Changes in TC statistics, recurving TC statistics, and Europe-impacting PTC statistics are considered in this study. A recurving
TC is defined as a TC which enters a domain in the North Atlantic midlatitudes from 82W-30E, 36-70N (as in Sainsbury et al.
2022). A Europe-impacting PTC is defined as a TC which enters a European domain defined as 10W-30E, 36-70N (as used in
Baker et al., 2021; Sainsbury et al., 2020, 2022b). The regions are constructed such that every Europe impacting PTC is a
recurving TC by definition and are illustrated in Figure 1. PTCs can reach Europe with either a cold core or a lower level warm
core (warm seclusion development, Baker et al. (2021); Dekker et al. (2018)). In this study, both types are considered.
In section 3.2.2, we investigate North Atlantic TC genesis regionally. North Atlantic TCs are separated based on genesis into
three regions: the Main Development Region (MDR, 0-20N, 70W-30E), Subtropical Atlantic (SUB, 20-30N, 82W-30E) and
the western Atlantic comprising the Gulf of Mexico and the region south of the Caribbean (denoted WEST, (4N,70W),
(4N,90W), (14N,90W), (14N,100W), (30N,100W), (30N,82W), (20N,82W) and (20N,70W)). These regions span the entire
tropical North Atlantic, and all simulated TCs form in one of these regions.

**2.5. Environmental field analysis**

Changes in large-scale environment fields known to be associated with TC genesis and intensification are investigated in
CMIP6 using the genesis potential index (GPI, Emanuel and Nolan (2004)).

$$GPI = |10^5 \eta|^{3/2} \left(\frac{H}{50}\right)^3 \frac{V_{pot}}{70} (1 + 0.1 V_{shear})^{-2},  \tag{1}$$

where $\eta$ is the 850hPa absolute vorticity, $H$ is the relative humidity at 600hPa, $V_{shear}$ is the magnitude of the 250-850hPa wind
shear and $V_{pot}$ is the potential intensity (PI) (Emanuel, 1986), implemented using the tcPyPI python package (Gilford, 2021).
Fields are first re-gridded to a common resolution (1x1 degrees) to ensure comparability.
Deep layer steering flow is also used and is defined as in Colbert and Soden (2012). However, due to data availability, we use
the 250hPa field instead of the 200hPa field.



## 3. Results

### 3.1. North Atlantic tropical cyclone climatologies in historical CMIP6 simulations

In this section we examine the climatology and properties of the TCs simulated by each of the selected CMIP6 models. While CMIP6 models generally have a higher resolution than CMIP5, their resolution is still lower than needed to simulate moderate and strong TCs explicitly (Davis, 2018). If we wish to learn something about how PTC impacts may change across Europe in the future, we need to understand whether these models are able to capture features of the observed TC climatology, and also identify any biases which may translate into biases in the projected changes in PTC statistics. Spatial patterns of TC genesis and the seasonal cycle are compared to those found in ERA5 over the same time period (1984-2014) using a consistent tracking and TC identification method. Due to biases in ERA5 TC intensity, further analysis of the TC lifetime maximum intensity (LMI) and wind-pressure relationship in the CMIP6 models is compared with observational data from HURDAT2 over the same time period.

### 3.1.1. Spatial statistics

Figure 1 shows the genesis density in the historical period for the five selected CMIP6 models (a-e) and ERA5 and HURDAT2 (f). Comparisons between HURDAT2 and ERA5/CMIP6 models should be made cautiously due to differences in how TCs are identified. The CMIP6 models, ERA5 and HURDAT2 typically show two regions of genesis maxima: one centred between 0 and 30W in the eastern tropical Atlantic, where the African Easterly Waves that act as TC precursors originate (Arnault and Roux, 2011; Thorncroft and Hodges, 2001), and a second to the western side of the basin. In ERA5 and HURDAT2, TC genesis is more of a continuum across the tropical Atlantic, whereas in the CMIP6 models (with the exception of CNRM) TC genesis is in two discrete regions. CNRM and HadGEM most closely match the spatial distribution of genesis seen during the same period in ERA5 and HURDAT2. Further information on the proportion of North Atlantic TCs forming in different subregions of the basin in CMIP6 and ERA5 can be found in Table 3.

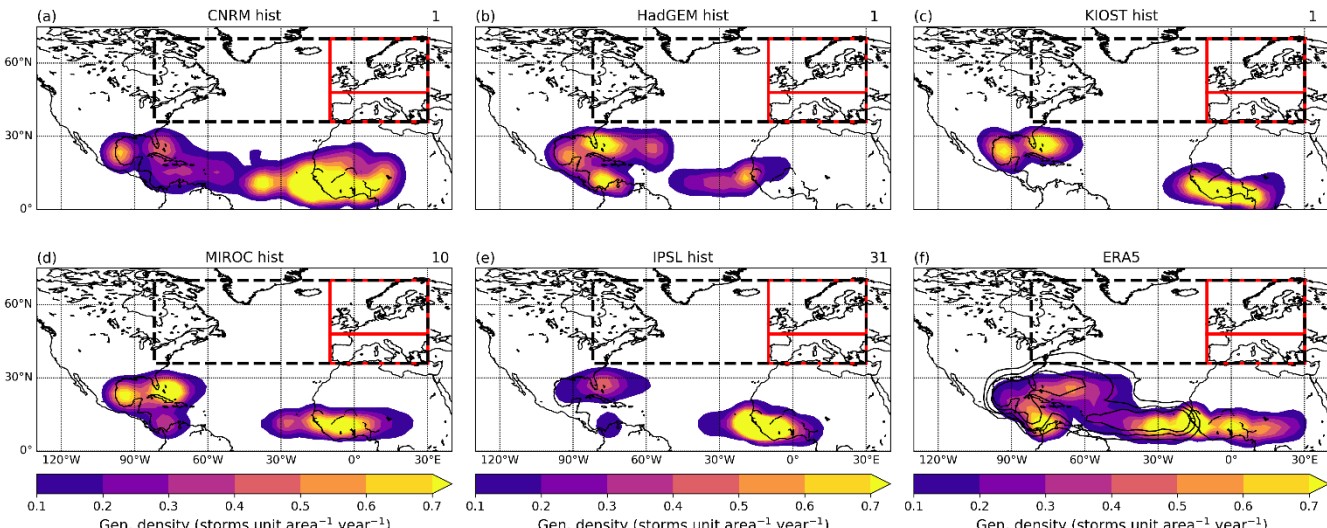

**Figure 1: Genesis density (storms per unit area per year, where the unit area is equal to a spherical cap with a 5-degree radius) for the 1984-2014 period from the historical runs of 5 CMIP6 models (a-e), and ERA5 (filled) and HURDAT2 (black lines) (f). Only TCs forming during the North Atlantic hurricane season are considered. The number of ensemble**
190 **members used is shown to the top right of each panel. Densities of less than 0.1 have been masked for clarity. HURDAT2 contours in (f) are 0.1, 0.3, 0.5 and 0.7 storms unit area$^{-1}$ year$^{-1}$, where the unit area is equal to a spherical cap of 5-degree radius. Black dashed region represents the recurvature domain, red regions represent the Northern Europe (top), Southern Europe (bottom) and Europe (whole red region) domains.**

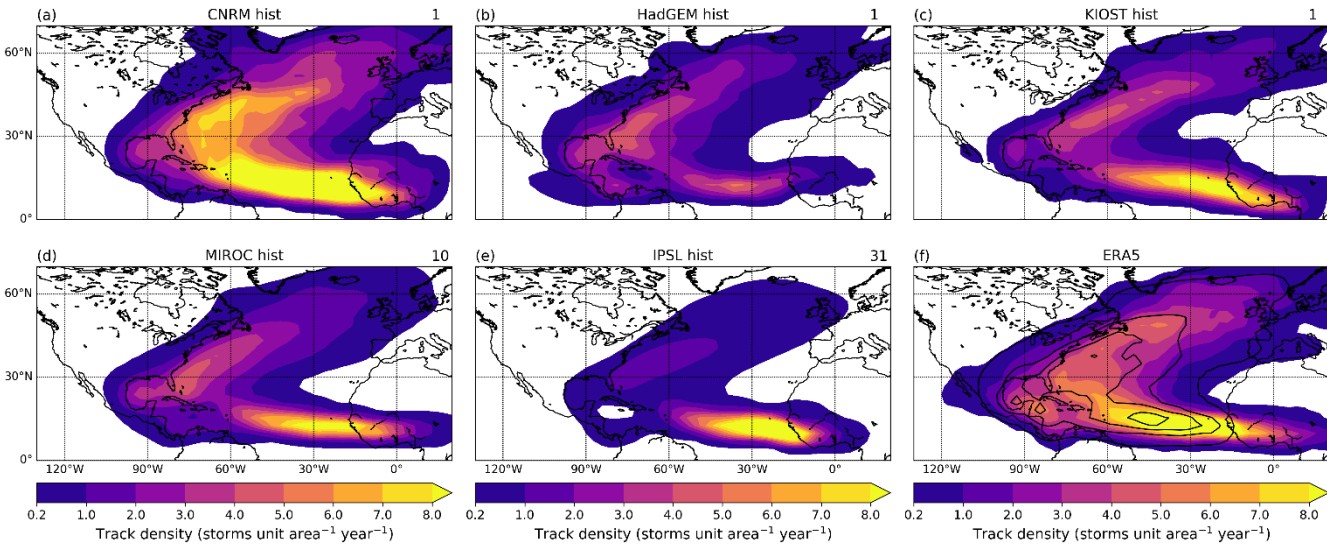

195 **Figure 2: As in Figure 1, but for track density. Densities less than 0.2 have been masked for clarity. Black contours in (f) represent 0.2, 2.0, 4.0, 6.0 and 8.0 storms unit area$^{-1}$ year$^{-1}$.**





All of the models capture the maxima in track density in the main development region (MDR) and the maxima in track density recurving around the US East Coast, heading towards Europe (as shown in Baker et al. (2021)). As with genesis density (Figure 1), many of the models have two apparent storm tracks, one associated with storms originating in the MDR and one associated with storms originating further west in the basin. In all models except CNRM, track density decreases rapidly from east to west across the MDR, and this is particularly clear in IPSL (Fig. 2e). The lysis density (Figure S3) is greater in the MDR in KIOST, MIROC, and IPSL than it is in ERA5, indicating that in these models TCs forming in the MDR dissipate too quickly. This is particularly clear for IPSL, which shows almost all MDR TCs dissipating whilst still in the MDR, close to where they formed.

Coupled with the lack of genesis in the western MDR in these models (Figure 1), the track (Figure 2) and lysis (Figure S3) density plots suggests that conditions in the models are too hostile for TC genesis or intensification in this region. In particular, vertical wind shear in all models except CNRM is higher (~2-6ms$^{-1}$) than ERA5 over the 1984-2014 period, with the biggest biases in the central and western MDR (Figure S4), consistent with previous studies (Han et al., 2022). Another possibility is that the conversion rate of seeds to TCs, which is sensitive to resolution, is too low in these models.

### 3.1.2. Seasonal cycle

Figure 3 shows the seasonal cycle for the selected CMIP6 models and ERA5. TCs in HURDAT2 are identified later in their lifecycle than TCs tracked and identified objectively (section 2.2, 2.3) in ERA5 and CMIP6 models. HURDAT2 data is therefore not included in Figure 3. CNRM has a bias towards early season North Atlantic TC activity (compared to ERA5), with a peak in August. This can also be seen in KIOST, but to a lesser extent. The other three CMIP6 models all show a peak in North Atlantic TC formation in September, the same as in ERA5. While the seasonal cycle is captured well by the models, all but CNRM underestimate North Atlantic TC frequency during hurricane season, with the largest underestimation found in HadGEM during the months of peak activity (August-October), consistent with too few simulated TCs originating in the MDR (Figs 1, 2). All models except the CNRM underestimate Europe-impacting PTC frequency (Fig. 3b) towards the beginning of the season (June – September).



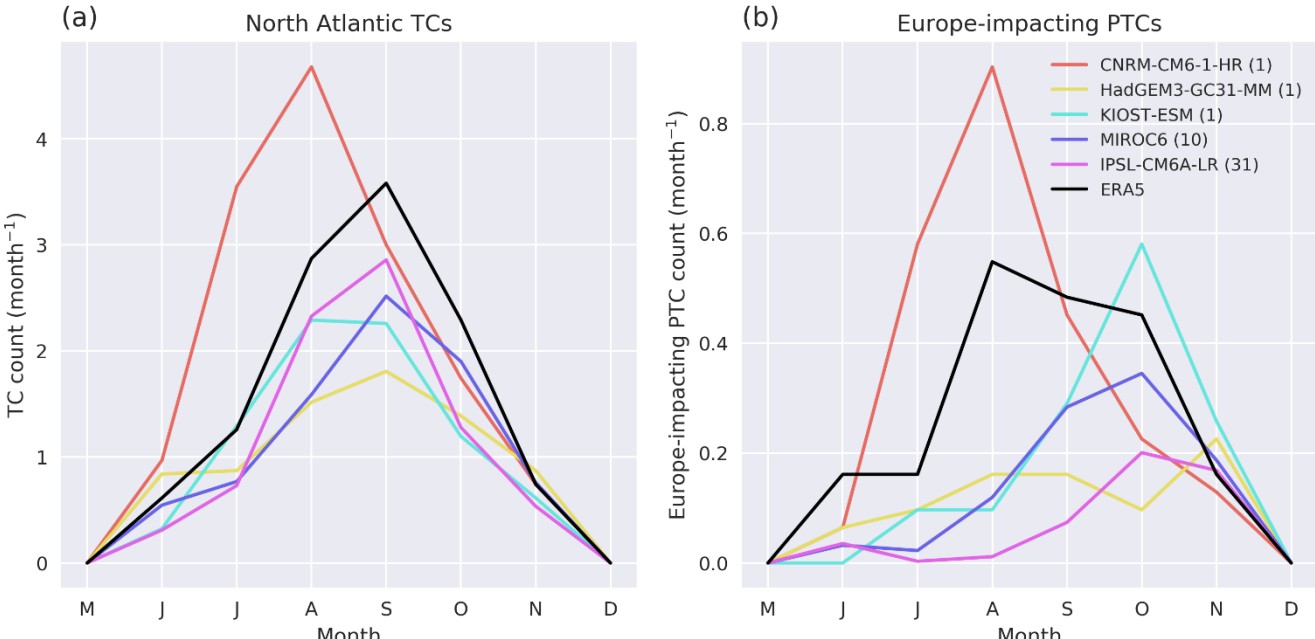

**Figure 3: Seasonal cycle of North Atlantic TCs (a) and Europe-impacting PTCs (b) in the historical run (1984-2014) of 5 CMIP6 models (coloured lines) and ERA5 (same time period, black). Cyclones are binned by the month of genesis. TCs forming outside of hurricane season are not included in this study and so these months are set equal to zero. The number of ensemble members used for each model is shown in brackets in the key.**

### 3.1.3. Lifetime Maximum Intensity

TC lifetime maximum intensity (LMI) distributions for the selected CMIP6 models are shown in Figure 4. All models use the same sampling frequency for wind speed (3hrPt). All selected CMIP6 models underestimate the mean TC LMI and are unable to simulate the strongest observed TCs. CNRM is able to simulate stronger TCs than the other CMIP6 models and ERA5, with some TCs approaching 50ms$^{-1}$ (Category 3 on the Saffir-Simpson scale). MIROC is able to simulate storms with a similar intensity to ERA5, however is still biased towards weaker TCs. The other CMIP6 models are unable to simulate TCs with LMIs of hurricane force (33ms$^{-1}$). The wind-pressure relationship (Fig. S5) also shows that TCs in ERA5 and CMIP6 models have wind speeds too low for a given sea level pressure (compared to HURDAT2).

The majority of TCs identified in the historical period of IPSL are extremely weak, with 10m wind speeds less than tropical storm (17ms$^{-1}$) strength. The large peak in TC LMI between approximately 10 and 15ms$^{-1}$ in IPSL corresponds to TCs forming in the MDR. TC LMI values in the right-hand tail of the IPSL distribution correspond to TCs originating in the Gulf of Mexico and along the Gulf Stream (not shown). The TCs in IPSL forming in the Gulf of Mexico and along the Gulf stream are forming at higher latitudes (~25-30N) than those in the MDR (~10-20N). One possibility is that TCs forming in the deep tropics (MDR) are purely diabatically driven, whereas those forming between 25 and 30N derive a component of their energy from baroclinic




240     sources (consistent with Elsner et al. (1996); Kossin et al. (2010)). Any issue with the parametrisation of diabatic fluxes in IPSL would therefore lead to MDR TCs which are much too weak but would not affect higher-latitude forming TCs as strongly, potentially explaining the difference. IPSL also uses a regular horizontal grid (Boucher et al., 2020) and so effective resolution increases with latitude. Systems are likely to be larger in scale at higher latitudes, and hence better represented at this resolution than at lower latitudes (for example, in the MDR). This may also in part explain the better representation of TC intensity with

245     latitude.

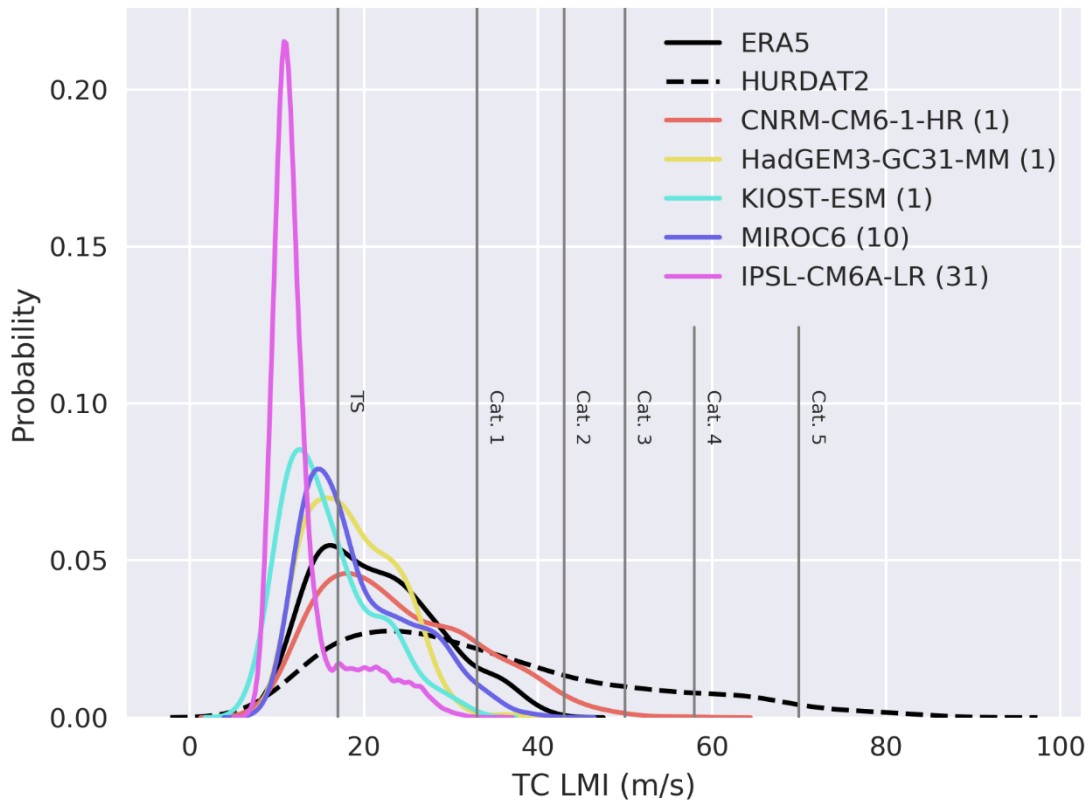

**Figure 4. TC lifetime maximum intensity distributions for the historical (1984-2014) period for selected CMIP6 models, ERA5 (black, solid), and HURDAT2 (black, dashed). Only TCs forming during the North Atlantic hurricane season are considered. Densities are approximated as kernel density estimates. Vertical grey lines represent the different**

250     **categories on the Saffir-Simpson scale. The number of ensemble members used for each model is shown in brackets in the key.**



Despite clear model biases, the selected CMIP6 models represent many features of the observed TC climatology, with spatial patterns and frequencies in qualitative agreement with observations. TC frequency, seasonal cycle, and spatial distribution in these selected CMIP6 models are comparable to those found in higher-resolution modelling studies, such as Climate-SPHINX (Vidale et al., 2021), UPSCALE (Roberts et al., 2015) and HighResMIP-PRIMAVERA (Roberts et al., 2020a; Haarsma et al., 2016; Baker et al., 2022), which used the same tracking and identification scheme. However, as expected, TC intensities are considerably lower in the CMIP6 models than identified in high-resolution studies.

### 3.1.4. Recurring TC and Europe PTC statistics

To gain insight into the projected changes in Europe-impacting PTCs, CMIP6 models must also capture the key features of the recurving TC, and Europe-impacting PTC climatologies. Previous work has shown that, to first order, TC activity governs recurving TC frequency (Sainsbury et al., 2022a). The selected CMIP6 models also capture the strong relationship between TC frequency and recurving TC frequency (Figure S6), highlighting that the models can capture the main driver of the interannual variability of recurving TCs, which may have important implications for Europe PTC risk.

A key feature of the observed PTC climatology is that PTC maximum intensities over Europe are on average, larger than those found for the broader class of midlatitude cyclones (MLCs, defined as all cyclones which are not PTCs) forming during hurricane season (Sainsbury et al., 2020). In Figure 5, we identify the maximum intensity associated with each PTC and MLC over Europe and subregions (northern/southern Europe, shown in Fig. 1) and investigate the fraction of cyclones in each intensity bin which are PTCs.



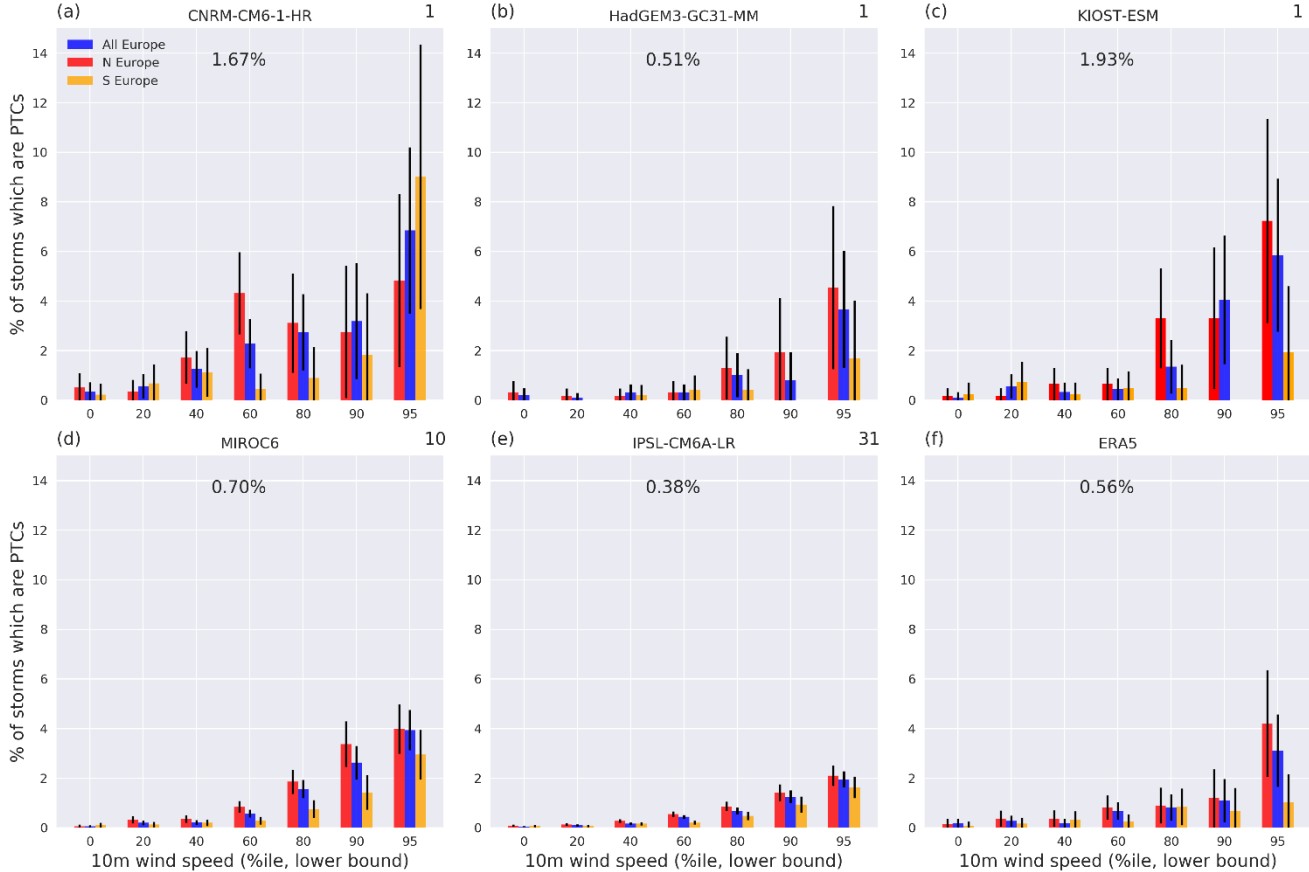

**Figure 5: Fraction of hurricane season forming, Europe-impacting cyclones which are PTCs for CNRM (a), HadGEM (b), KIOST (c), MIROC (d) and IPSL (e), and ERA5 (f), where cyclones are binned by their maximum 10m wind speed over Europe (blue), Northern Europe (red) and Southern Europe (yellow) in the historical runs of the selected CMIP6 models (a-e). Northern Europe is defined as 48-70N, 10W-30E, and southern Europe is defined as 36-48N, 10W-30E. Percentage of cyclones impacting Europe (whole domain) during the North Atlantic hurricane season which are PTCs is shown on each panel. The number of ensemble members used for each model is show to the upper-right of each panel. Vertical bars show the 95% binomial proportion confidence interval.**

To ensure that the sample size remains reasonable across bins and across models, we bin the cyclones based on percentiles of the combined distribution. For each model, we combine the (Europe-impacting) PTC and MLC cyclone tracks over both time periods (historical and future) and calculate the $20^{th}$, $40^{th}$, $60^{th}$ $80^{th}$, $90^{th}$ and $95^{th}$ percentiles of this joint distribution of their maximum 10m wind speeds over Europe. These percentiles are then used to bin the data.

In ERA5, only 0.56% of cyclones reaching Europe during the North Atlantic hurricane season are PTCs. However, when considering the highest intensity bin ($>95^{th}$ percentile), this fraction is 3.11%, over 5 times larger. While these numbers are



different to those found in Sainsbury et al. (2020) due to differences in date range and the bins used to bin the data, the key point remains: There is an increasing trend in the fraction of cyclones which are PTCs with intensity. Despite being unable to simulate intense TCs, all five CMIP6 models capture the relationship between TC frequency and recurving TC frequency, and the disproportionate intensity associated with Europe-impacting PTCs.

### 3.2. Projected changes in Europe-impacting PTC frequency

In this section we investigate the projected changes in Europe-impacting PTC counts. We consider the projected changes in three key components: i) Changes in basin-wide North Atlantic TC counts, ii) Changes in the likelihood that a North Atlantic TC will recurve, and iii) Changes in the likelihood that a recurving North Atlantic TC will reach Europe.

While some overlap exists, these three components are likely driven by different factors. Changes in basin-wide TC counts to an extent depend on how the large-scale environment (sea surface temperature, vertical wind shear, atmospheric moisture, etc.)
and teleconnections (e.g., ENSO) change in the future (e.g., genesis potential index, Camargo (2013)). Changes in likelihood of recurvature may depend on changes to the large-scale steering flow, changes in TC intensity (stronger TCs survive longer), changes in where TCs are forming (TCs in some regions are more prone to recurve than in other regions, Sainsbury et al. (2022)), and changes to the large-scale environmental conditions in the subtropical Atlantic (more favourable conditions for TCs in the subtropics may lead to a larger proportion of TCs successfully making the transit from the tropics to the extratropics
(Haarsma et al., 2013)). Changes in the likelihood that a recurving TC will reach Europe may be related to changes in the midlatitude jet and the intensity of TCs (Haarsma, 2021; Sainsbury et al., 2022b).

Europe-impacting PTC counts are therefore expressed as

$$N_{Europe} = N_{TC}F_{Rec}F_{Europe\,|\,Rec} \,, \tag{2}$$


where $N_{Europe}$ is the number of Europe-impacting PTCs, $N_{TC}$ is the number of North Atlantic TCs, $F_{Rec}$ is the fraction of North Atlantic TCs which recurve (the likelihood of recurvature), and $F_{Europe\,|\,Rec}$ is the fraction of recurving TCs which reach Europe (the likelihood that a recurving TC will reach Europe). Each term is calculated for the historical and future SSP5-85 projection for each model and shown (with the fractional changes) in Table 2.

All five selected models project a statistically significant (to 95%) decrease in North Atlantic TC frequency ($N_{TC}$) of 30-60% by the end of the 21st century. Four models also show an increase in the likelihood of recurvature ($F_{Rec}$), which is significant in HadGEM, MIROC and IPSL. HadGEM is the only model with a significant projected increase in $F_{Eur|Rec}$, and all other models have non-significant projected changes. Overall, there is no robust model response in Europe-impacting PTC frequency in the future ($N_{Europe}$), with CNRM projecting a significant decrease, MIROC projecting a significant increase, and the
remaining models showing no significant change.



In all models except CNRM, the fractional decrease in TC counts is much larger than the fractional change in Europe-impacting PTC counts, with two models even projecting an increase in Europe-impacting PTC counts in the future. Therefore, in four of the five models, there is a projected increase in the proportion of North Atlantic TCs which reach Europe in the future ($=F_{Rec}\,F_{Eur|Rec}$), which is significant in MIROC and IPSL (Fig. 10a). In HadGEM, IPSL and MIROC the reduction in TC
counts is offset by a projected increase in the likelihood of recurvature, highlighting that future TCs may be more likely to impact the heavily-populated US East Coast. The projected increase in $F_{rec}$ is consistent with the projected increase in potential intensity and projected decrease in vertical wind shear along the US East Coast (Figure 6). This result also supports previous studies which suggest that more favourable conditions in the subtropical North Atlantic will allow more TCs to survive the transit from the tropics to the extratropics in the future (Haarsma et al., 2013). The projected changes shown in Table 2 are not
sensitive to whether all ensemble members are used, or whether only ensemble members common to both the historical and future periods (same realization, initialization, and physics) are used (Table S1).

|  | $N_{TC}$ | | | $F_{rec}$ | | | $F_{Europe|rec}$ | | | $N_{Europe}$ | | |
|---|---|---|---|---|---|---|---|---|---|---|---|---|
|  | Hist | SSP | Diff | Hist | SSP | Diff | Hist | SSP | Diff | Hist | SSP | Diff |
| CNRM | 14.68 | 9.00 | **-39%** | 0.55 | 0.59 | 7% | 0.29 | 0.24 | -17% | 2.35 | 1.29 | **-45%** |
| HadGEM | 7.29 | 4.35 | **-40%** | 0.46 | 0.61 | **31%** | 0.24 | 0.35 | **49%** | 0.81 | 0.94 | 16% |
| KIOST | 7.97 | 5.19 | **-35%** | 0.41 | 0.40 | -4% | 0.40 | 0.50 | 24% | 1.32 | 1.03 | -22% |
| MIROC | 8.07 | 5.53 | **-31%** | 0.43 | 0.68 | **58%** | 0.29 | 0.31 | 9% | 0.99 | 1.17 | **18%** |
| IPSL | 8.04 | 3.16 | **-61%** | 0.20 | 0.42 | **110%** | 0.31 | 0.32 | 3% | 0.49 | 0.42 | -15% |

**Table 2: Counts of Europe-impacting PTCs, North Atlantic TCs, likelihood of recurvature, and likelihood that a**
**recurring TC will reach Europe for the historical (1984-2014) period and the future (2069-2099) period under the SSP5-85 scenario. Fractional changes shown under the 'Diff' columns. Bold values represent significance at the 95% level using a bootstrapping method.**

### 3.2.1. Projected change in the number of North Atlantic TCs ($N_{TC}$)

To investigate the significant projected decrease in North Atlantic TC counts (Table 2), the projected change in the genesis
potential index and its terms (as calculated in section 2.5) during the North Atlantic hurricane season are calculated and shown in Figure 6. Overall, the genesis potential index is projected to significantly increase along the US East Coast between approximately 30N and 40N (Fig. 6a-e). This indicates that in the future, TCs travelling through this region will be exposed to more favourable conditions for TCs, increasing the probability that they reach the recurvature domain. This is consistent with the projected increase in the likelihood of recurvature found in Table 2. In the selected models, the increase in GPI in this

region is associated with a projected increase in potential intensity (PI, row 4) and a decrease in vertical wind shear (VWS, row 5), significant across all models.

While the projected changes in Figure 6 are consistent with an increased probability of recurvature, they do not help to explain the significant decrease in basin-wide TC counts towards the end of the century. Previous studies have shown that saturation deficit may be a better metric for TC genesis potential than relative humidity (Emanuel et al., 2008) and is projected to increase in the future (increasing hostility). Furthermore, it has been proposed that an increase in static stability may lead to a reduction in TC frequency (Bengtsson et al., 2007; Sugi et al., 2002). These factors may help to explain why we see such a large projected decrease in TC counts in the North Atlantic despite an overall increase in the GPI.

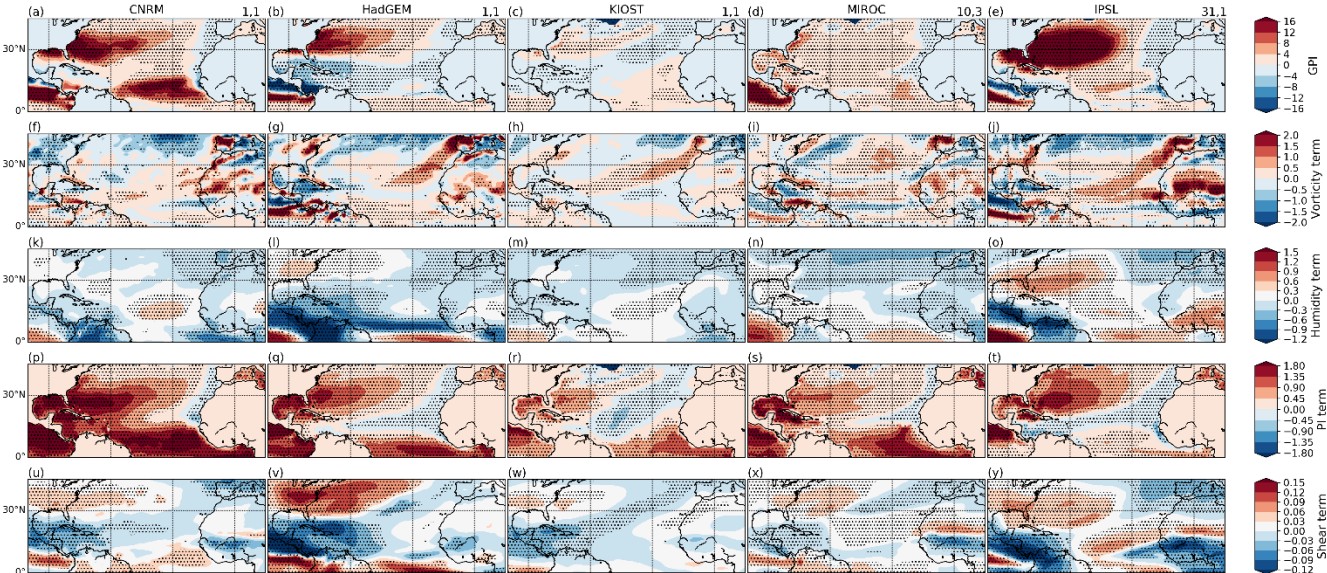

**Figure 6: Projected change (future minus historical) in the GPI (top row) and the individual terms of the GPI equations (rows 2-4) for each of the selected CMIP6 models: vorticity term (second row), humidity term (third row), PI term (fourth row) and shear term (fifth row). Note that the vertical wind shear term is a function of the reciprocal of the wind shear, and so a positive difference indicates less vertical wind shear in the future. Number of ensemble members used for the historical and future periods are shown to the upper-right of each column (historical, future). Stippling represents statistical significance at the 95% level using Welch's t-test.**

**3.2.2. Projected change in the fraction of recurring North Atlantic TCs ($F_{Rec}$)**

Table 2 shows a statistically significant increase in the likelihood of recurvature in 3 of the 5 models. Whether or not a TC recurves could depend on multiple factors (Sainsbury et al., 2022a): the location in which the TC forms, TC intensity (stronger TCs are more resilient to hostile conditions), and the steering flow (Colbert and Soden, 2012). In this subsection, we aim to investigate which of these factors – if any – are responsible for the projected increased in $F_{Rec}$.





Projected increases in GPI along the US East Coast are consistent with the increased likelihood of recurvature. In this region, a reduction in wind shear is collocated with a projected increase in PI (Fig. 6p-6y). This is consistent with CMIP5 models (Camargo, 2013) and indicates that future TCs traversing the US East Coast may retain TC-like conditions further poleward, increasing their likelihood of both making it to the midlatitudes (and being identified as recurving as a result), and potentially also reaching Europe. The projected increase in GPI along the US East Coast supports previous studies which suggest an

increase in the latitude of TC LMI and an overall expansion of the tropical genesis region (Kossin et al., 2014; Haarsma, 2021). However, all five models show an increase in GPI along the US East Coast, but only three models have a significant increase in $F_{Rec}$, indicating that other factors must also play a role.

Figure 7 shows the normalized TC track density for the historical and future periods, along with the difference (future minus historical). The track densities are normalized by dividing by the total number of TCs, so the differences show the geographical

redistribution of North Atlantic TCs rather than the change in total number ($N_{TC}$).

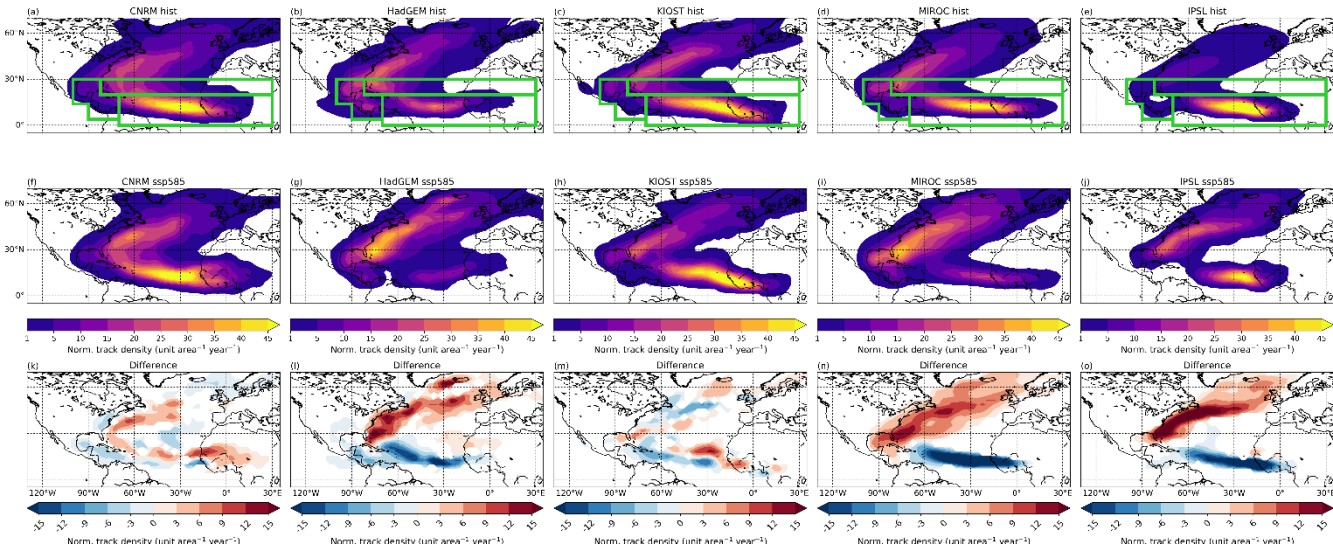

**Figure 7: Normalized TC track density for the 5 selected CMIP6 models during the historical (top) period, towards the end of the century under SSP5-85 (middle) and the difference (future minus historical, bottom). Densities less than 1**

**have been masked for clarity. Green domains represent the boundaries of the MDR, SUB and WEST regions.**

HadGEM, IPSL and MIROC have many similarities in normalized track density difference. There is proportionally less track density in the MDR and proportionally higher track density in the future along the East coast of the US heading towards Europe. The large decrease in normalized TC track density in the MDR indicates a potential shift in genesis, away from the

MDR towards the west of the North Atlantic, as confirmed by the normalized genesis densities (Fig. S8). To investigate this further, we separate the North Atlantic TCs based on genesis into three regions: the Main Development Region (MDR),





Subtropical Atlantic (SUB), and western Atlantic (WEST). These regions are constructed such that all North Atlantic TCs form in one of these regions, and the boundaries for these regions can be found in section 2.4.

We decompose the likelihood of recurvature based on these three regions of genesis:


$$F_{Rec} = W_{MDR}F_{MDR} + W_{SUB}F_{SUB} + W_{WEST}F_{WEST}, \tag{3}$$

where $W_i$ are weighting terms, representing the proportion of North Atlantic TCs which form in region i, and $F_i$ represents the proportion of TCs forming in region i which recurve (i=MDR, SUB, WEST). The 6 terms on the right-hand side of equation

3 are calculated for the historical and future runs of the five selected CMIP6 models and are shown in Table 3.

| HIST (1984-2014) | $W_{MDR}$ | $F_{MDR}$ | $W_{SUB}$ | $F_{SUB}$ | $W_{WEST}$ | $F_{WEST}$ |
|---|---|---|---|---|---|---|
| CNRM | 0.77 | 0.54 | 0.11 | 0.69 | 0.12 | 0.54 |
| HadGEM | 0.34 | 0.14 | 0.31 | 0.83 | 0.35 | 0.46 |
| KIOST | 0.62 | 0.17 | 0.18 | 0.91 | 0.19 | 0.73 |
| MIROC | 0.52 | 0.19 | 0.21 | 0.86 | 0.26 | 0.55 |
| IPSL | 0.75 | 0.03 | 0.14 | 0.83 | 0.10 | 0.58 |
| ERA5 | 0.62 | 0.46 | 0.19 | 0.75 | 0.19 | 0.33 |
| SSP5-85 (2069-2099) | $W_{MDR}$ | $F_{MDR}$ | $W_{SUB}$ | $F_{SUB}$ | $W_{WEST}$ | $F_{WEST}$ |
| CNRM | 0.76 | 0.54 | 0.16 | 0.78 | 0.08 | 0.68 |
| HadGEM | 0.17 | 0.22 | 0.42 | 0.74 | 0.41 | 0.64 |
| KIOST | 0.60 | 0.16 | 0.20 | 0.81 | 0.20 | 0.69 |
| MIROC | 0.25 | 0.32 | 0.33 | 0.90 | 0.41 | 0.72 |
| IPSL | 0.52 | 0.00 | 0.31 | 0.90 | 0.17 | 0.82 |
| **DIFF** | $W_{MDR}$ | $F_{MDR}$ | $W_{SUB}$ | $F_{SUB}$ | $W_{WEST}$ | $F_{WEST}$ |
| CNRM | -1% | 0% | **45%** | 13% | **-33%** | 26% |
| HadGEM3 | **-50%** | 57% | **35%** | -11% | 17% | **39%** |
| KIOST | -3% | -6% | 11% | -11% | 5% | -5% |
| MIROC | **-52%** | **66%** | **57%** | 4% | **56%** | **30%** |
| IPSL | **-31%** | **-100%** | **111%** | 8% | **71%** | **43%** |

**Table 3: Tabulated values of the terms of the right-hand side of equation (3) for the historical run (top), future run under SSP5-85 (middle) and the fractional change (bottom) for the five selected CMIP6 models. Bolded values**
**represent significant fractional changes at the 95% level using a bootstrapping method.**





Rows 2-6 of Table 3 highlight the significant bias historically of many of the models (all but CNRM) for recurvature of TCs originating in the MDR. Approximately 46% of MDR forming TCs recurve in ERA5, but this value is between 3% and 19% in four of the five models, with only CNRM correctly capturing this fraction. The three models which have a significant

increase in $F_{Rec}$; HadGEM, MIROC and IPSL, all see a significant shift in proportional genesis away from the MDR towards the SUB and WEST regions. Due to the $F_{MDR}$ bias in these models, the shift in genesis from the MDR to the other regions leads to an increase in $F_{Rec}$. A component of the projected increase in $F_{Rec}$ is therefore likely a manifestation of model biases. To quantify the contribution of genesis shifts to the projected change in $F_{Rec}$, we split the change in $F_{Rec}$ into three terms as described in the appendix (equation (A4)). Term 1 represents the contribution to the change in $F_{Rec}$ caused by a change in the

likelihood of recurvature within each region which recurve. Term 2 represents the contribution to the change in $F_{Rec}$ caused by a shift in genesis location, and term 3 represents the combination of these changes.

| Model | Term 1 | Term 2 | Term 3 |
|---|---|---|---|
| CNRM | 0.81 | 0.21 | -0.02 |
| HadGEM | 0.42 | **0.67 (0.28-1.81)** | -0.09 |
| KIOST | 1.84 | -0.96 | 0.11 |
| MIROC | **0.47 (0.31-0.59)** | **0.55 (0.47-0.67)** | -0.02 |
| IPSL | 0.07 | **0.77 (0.61-1.02)** | **0.16 (0.05-0.24)** |

**Table 4: Contribution to projected change in F$_{Rec}$ from terms 1 (second column), 2 (third column) and 3 (fourth column)**
**of Equation A4. Bolded values represent significance at 95% using a bootstrapping method. For significant values, the 95% confidence interval is shown in brackets.**

Term 2 dominates for the three models which have a significant increase in $F_{Rec}$, indicating a significant contribution to $F_{Rec}$ from a shift in genesis location away from the MDR to the SUB and WEST regions in these three (and only these three)

models. The projected change in GPI (Fig. 6) does not show a large increase in hostility in the MDR compared to other regions in the future. Changes in TC seeds have been shown to influence TC frequency (Vecchi et al., 2019), and so the shift in TC genesis away from the MDR could be the result of a projected decrease in the frequency or intensity of TC seeds in the MDR, or a change in the conversion rate of seeds into TCs. The results of Tables 3 and 4 are not sensitive to the exact position of the region boundaries (not shown).

While shifts in genesis explain most of the projected change in $F_{Rec}$ in HadGEM, MIROC and IPSL, they do not explain all of the projected increase. To further investigate the projected changes in $F_{Rec}$, the relationship between projected TC LMI changes and projected $F_{Rec}$ changes is explored in the MDR, WEST and SUB regions of the five selected models. Significant





increases in TC LMI are found in the WEST region of HadGEM, MIROC and IPSL, and in the MDR in MIROC. A significant increase in $F_{Rec}$ is also found in the same regions and models (Table 3). A significant relationship (Pearson's correlation coefficient of 0.52) is found between TC LMI changes and $F_{Rec}$ changes across the five models (Figure S9), suggesting that projected increases in TC LMI may be associated with the projected increases in $F_{Rec}$ in the HadGEM, MIROC and IPSL.

Figure 8 shows the change in the hurricane-season mean deep layer steering flow (Colbert and Soden (2012)). All models have a significantly weaker westerly flow between 30 and 40N over the US East Coast corresponding to the region in which shear decreases in the future (Figure 6). Differences in the steering flow are very small in the tropics in all models except HadGEM. In HadGEM, the easterly flow in the western tropical Atlantic is reduced, with increased poleward flow in the subtropical North Atlantic. This difference in flow between the historical and future periods would suggest an increased likelihood for recurvature for TCs forming at a given latitude in the WEST region (as the TCs are not being steered as strongly westwards towards land), and an increase in the likelihood of recurvature in the MDR, where the slower easterly flow in the western MDR and stronger poleward flow in the subtropics may aid recurvature. This is consistent with Table 3, which shows a significant increase in the likelihood of recurvature in the WEST region and a (non-significant) increase in the MDR.

Three of the five CMIP6 models project a significant increase in $F_{Rec}$. Section 3.2.2 suggests that a shift in genesis from the MDR towards the SUB and WEST regions is responsible for the majority of this projected increase, some of which is likely to be associated with historical model biases. In these three models, increases in TC LMI may also be associated with the projected increase in $F_{Rec}$, and in HadGEM, changes in the steering flow may also play a role. Reduction in vertical wind shear (VWS) and increases in PI (Figure 6), seen across the models, are also consistent with enhanced longevity, implying increased likelihood of recurvature.

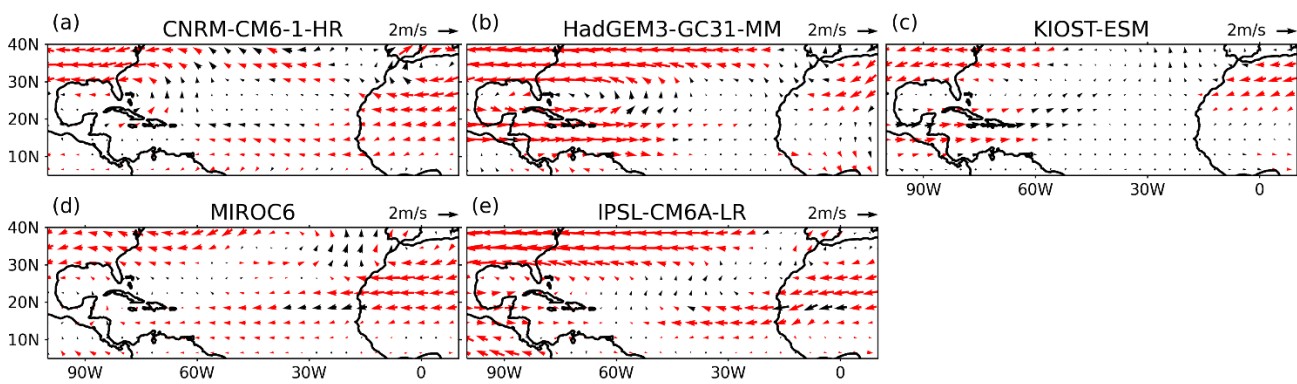

**Figure 8: Difference (future minus historical) in the hurricane-season averaged deep layer steering flow for the five selected CMIP6 models. Statistically significant differences between the future and historical period at the 95% level are shown in red and are calculated using Welch's t-test.**





### 3.2.3. Projected change in the fraction of recurving North Atlantic TCs which reach Europe ($F_{Eur|Rec}$)

While 4 of the 5 CMIP6 models agree on the sign of the change in $F_{Eur|Rec}$, this projected increase is only significant in one
model, HadGEM. This is associated with a shift in seasonality in HadGEM. In the future in this model, most recurving TCs interact with the midlatitudes later in the hurricane season, a result unique to HadGEM (not shown). A similar change in seasonality of extratropical transition was found across HighResMIP models (Baker et al., 2022). Climatologically, midlatitude baroclinicity increases throughout hurricane season and so in the future, many recurving TCs in HadGEM encounter a more favourable midlatitude environment. This suggests that recurving TCs in HadGEM in the future are more likely to undergo
extratropical reintensification, which has been shown to be linked to whether a recurving TC will reach Europe (Sainsbury et al., 2022b).

### 3.3. Projected changes in Europe-impacting PTC intensity

In this section we investigate how the intensity of PTCs may change by the end of the century. Figure 9 shows the absolute
number (per ensemble member) of Europe-impacting PTCs in each bin during the historical and future periods (bars), with the fractional change overlaid. CNRM and KIOST show both a decrease in the absolute number of strong Europe-impacting PTCs (Fig. 9a, e). IPSL and MIROC ensembles show an increase. HadGEM is mixed, with a decrease in the number of PTCs in the highest intensity bin but increases in the second and third highest intensity bins (Fig. 9b). The projected changes in Europe-impacting PTC intensity shown in Figure 9 are not significantly different if reproduced using only ensemble members common
to both the historical and future periods (Figure S7).

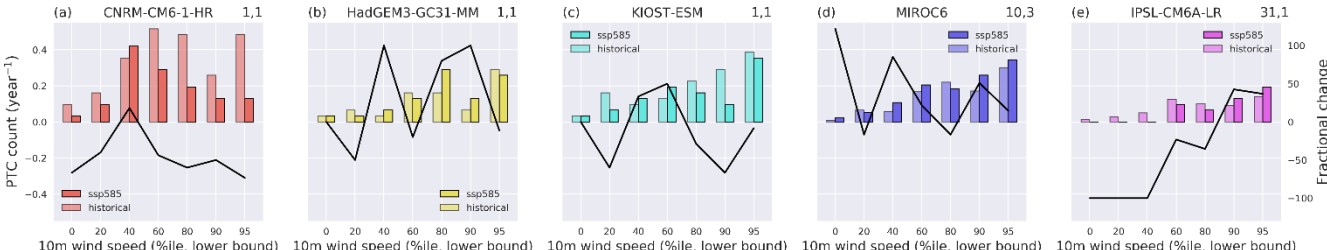

**Figure 9: Bar plot showing the number of Europe-impacting PTCs (per ensemble member) in each intensity bin for the five selected CMIP6 models for the historical (1984-2014, lighter colours) period and towards the end of the century**
**under the SSP5-85 scenario (2069-2099, darker colours). Fractional change in the counts in the future period compared to historical is shown as the black line corresponding to the right-hand side Y axis. Number of ensemble members used for the historical and future periods are shown to the upper right of each panel (historical, future).**



In HadGEM, KIOST, MIROC and IPSL, the decrease in TC frequency basin-wide is considerably larger than the change in
strong Europe-impacting PTCs. For example, MIROC has an increase in the number of strong Europe-impacting PTCs despite
a 31% reduction in the number of North Atlantic TCs. This implies that the proportion of North Atlantic TCs which impact
Europe as strong PTCs is projected to increase. This is illustrated in Figure 10, which shows the proportion of all North Atlantic
TCs which reach Europe as strong PTCs (Fig. 10b) and very strong PTCs (Fig. 10c). Strong PTCs are defined as PTCs which
impact Europe with winds greater than the 90$^{th}$ percentile of the distribution of maximum winds over Europe during hurricane
season (considering all PTCs and MLCs in the historical and future period). Very strong PTCs are PTCs which impact Europe
with winds greater than the 95$^{th}$ percentile.

Four of the five models show an increase in the proportion of North Atlantic TCs which reach Europe as strong and very strong
PTCs, and this difference is statistically significant in IPSL and MIROC. Our results therefore suggest that the future risk
posed by PTCs to Europe may depend on how TC activity basin-wide changes in the future. If TC frequency decreases
substantially (as suggested by this analysis), then the number of strong Europe-impacting PTCs is unlikely to change
significantly. However, if TC frequency does not decrease much, or potentially increases, then Europe could be subject to
significantly more strong PTCs in the future, as was found in Haarsma et al (2013).

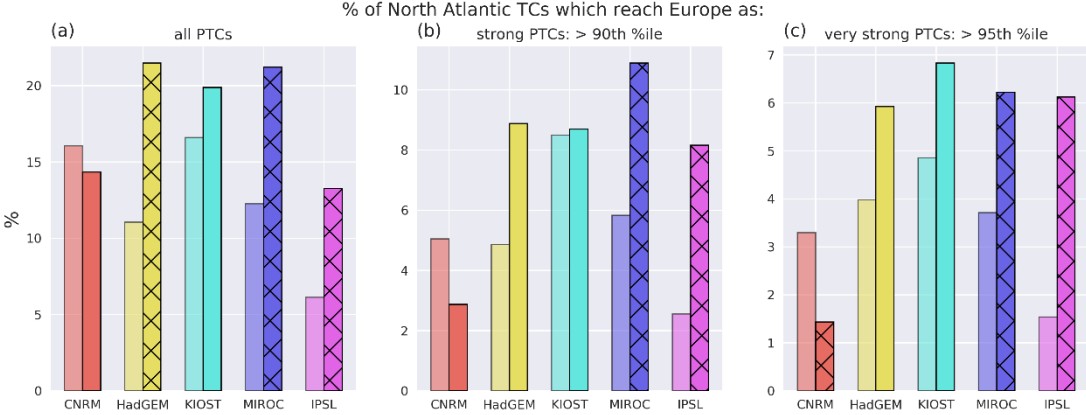

**Figure 10: Bar charts showing the proportion of North Atlantic TCs which impact Europe as (a) PTCs, (b) strong PTCs
(max winds > 90$^{th}$ percentile of the distribution of maximum winds over Europe) and (c) very strong PTCs (>95$^{th}$
percentile). Lighter colours show the values for the historical period, darker bars for the future period. Hatching is
overlaid in models where the projected change is significant to 95% using a bootstrapping method.**

Haarsma et al. (2013) find a large increase in the frequency of hurricane force PTCs reaching Europe by the end of the century.
The interpretation of Figures 9 and 10 does not change when using the regions (Norway, North Sea, West UK & Ireland, and
Bay of Biscay) and season (August-October) used in Haarsma et al. (2013) (Figs S10-S12). Despite using RCP 4.5, the
prescribed SSTs used in Haarsma et al. (2013) are similar to the projected SST changes found in this study (not shown). The



differences between Figure S10 and Figure 2f in Haarsma et al. (2013) could be caused by different projected changes in North
Atlantic TC counts (which were not investigated in their study), differences in model resolution, differences in TC
identification methodology, or differences model configuration (coupled vs atmosphere only).

## 4. Discussion and Conclusions

In this study, we have presented the first multi-model analysis of how Europe-impacting PTC frequency and intensity may
change by 2100. Using a vorticity-based tracking scheme and objective TC identification method, we identify all North Atlantic
TCs in five CMIP6 models in the historical (1984-2014) period, and the future (2069-2099) period under the SSP5-85 scenario,
using all available ensemble members. These five models were selected from a wider sample of CMIP6 models based on their
ability to simulate North Atlantic TC frequency compared to observations (Figure S1). While CMIP6 models do not have
sufficient resolution to resolve all TC-related processes, the number of models and ensemble members allows us to investigate
projected Europe-impacting PTCs changes with a considerably larger TC sample size than available for previous studies. The
key results are as follows:

- The five selected CMIP6 models are able to simulate many aspects of the North Atlantic TC climatology compared
  to observations. They capture the relationship between TC frequency and recurving TC frequency, and capture the
  disproportionate risk associated with PTCs compared to extratropical cyclones over Europe. However, the models
  still have many deficiencies. In particular, TCs forming in the MDR are too short lived and therefore unlikely to
recurve, and TC intensity is significantly underestimated.

- No robust model response in Europe-impacting PTC frequency (overall, or as strong storms) is found in the future.
  This is because two competing factors – a decrease in North Atlantic TC frequency, and an increase in the
  proportion of TCs reaching Europe – are of approximately the same size.

- The projected decrease in North Atlantic TC frequency is statistically significant in all five models, with decreases
of between 30 and 60% found by the end of the 21$^{st}$ century.

- The projected increase in the proportion of TCs reaching Europe is found in four of the five models and is
  associated with a projected increase in the likelihood of recurvature. The increased likelihood of recurvature may be
  associated with a more favourable environment for TCs along the US East Coast, where wind shear is projected to
  decrease, and potential intensity is projected to increase in the future. This result is also consistent with previous
studies which highlight that conditions between where TCs typically form and Europe are overall likely to become
  more favourable for tropical cyclogenesis in the future (Haarsma et al., 2013; Baatsen et al., 2015; Liu et al., 2017).

- The projected increase in the likelihood of recurvature in the North Atlantic is also associated with a shift in
  genesis, with proportionally less TCs forming in the MDR in the future, where model biases cause very few TCs to
  recurve.



Our results highlight the large uncertainty associated with projected changes in Europe-impacting PTC intensity and frequency. Even the model with the largest projected increase in intense Europe-impacting PTCs has a considerably lower increase than found in previous studies (Haarsma et al., 2013; Baatsen et al., 2015). The large uncertainties in the projected responses are anticipated – model uncertainties in TC genesis (Yamada et al. 2021; Yang et al. 2021; Vecchi et al. 2019; Camargo 2013; Ting et al. 2015), TC recurvature (Colbert and Soden, 2012), TC intensity (Kossin et al., 2020) and midlatitude environment

(for example, jet location and intensity (Harvey et al., 2020)) could translate to model uncertainty in Europe-impacting PTCs due to the complex lifecycle of these systems.

Projected decreases in North Atlantic TC counts are found in many previous studies which explicitly track TCs (Roberts et al., 2015; Gualdi et al., 2008; Rathman et al., 2014), but not all (e.g., Bhatia et al. 2018). There are also physical arguments which support a decrease in TC activity due to an increase in static stability (e.g., Bengtsson et al. (2007); Sugi et al. (2002)). However,

other methods such as statistical and dynamical downscaling are more mixed in terms of the sign of the projected change (Emanuel, 2021, 2013; Jing et al., 2021), and there are often sensitivities to the tracking scheme when TCs are tracked explicitly (Roberts et al., 2020b). Previous studies have also suggested a broadening of weak TC circulations in the future (Sugi et al., 2020), which would result in future TCs having lower associated vorticity. As a result, tracking schemes which used a fixed vorticity threshold may capture a lower proportion of all model-simulated TCs in the future. The use of a percentile-based

vorticity threshold may alleviate this problem. It is therefore necessary to reduce the uncertainty associated with North Atlantic TC frequency projections before greater confidence in future European PTC risk can be achieved. This should involve further work on our theoretical understanding of what drives TC genesis, and further quantification of the uncertainty associated with different TC identification methods (e.g., Bourdin et al. 2022).

Model biases, particularly in the MDR, are likely to manifest in the future projections. Furthermore, TC LMI, which is not

adequately captured by these models, has been shown to be associated with the likelihood of recurvature (Sainsbury et al., 2022a) and the likelihood that a recurving TC will reach Europe (Sainsbury et al., 2022b). The model deficiencies in TC intensity may therefore be contributing to the low bias in likelihood of recurvature across many of the models during the historical period. Therefore, CMIP6 models must be used cautiously when investigating projected changes to TC recurvature or Europe PTC impacts in the future. Previous studies suggest that TCs will be more intense in the future (Knutson et al., 2010,

2019; Bhatia et al., 2018; Bender et al., 2010; Emanuel, 2021; Walsh et al., 2019), implying greater longevity and a greater probability of reaching Europe (Sainsbury et al., 2022b). Multi-model studies using high-resolution climate models, which are capable of better simulating the distribution of TC intensities, are therefore necessary to fully explore the projected changes in Europe-impacting PTCs.

## 5. Appendix

By splitting the North Atlantic basin into different spatial regions, the fraction of recurving North Atlantic TCs in the historical, H, and future, S, periods can be defined as:



$$F^H = \sum_i W_i^H F_i^H \;,\;\; F^S = \sum_i W_i^S F_i^S, \tag{A1}$$

where $i$ = MDR, SUB and WEST, $W_i$ is the proportion of North Atlantic TCs forming in region $i$, and $F_i$ represents the fraction of TCs forming in region $i$ which recurve. The absolute change in the fraction of recurving TCs, $\Delta F = F^S - F^H$, can then be expressed as

$$\Delta F = \sum_i (W_i^S F_i^S - W_i^H F_i^H). \tag{A2}$$

Replacing $W_i^S$ with $W_i^H + \Delta W_i$ and $F_i^S$ with $F_i^H + \Delta F_i$, rearranging and cancelling common terms allows $\Delta F$ to be expressed as three separate terms

$$\Delta F = \sum_i W_i^H \Delta F_i + \sum_i F_i^H \Delta W_i + \sum_i \Delta W_i \Delta F_i. \tag{A3}$$

The relative contribution of each term can be investigated by dividing the right-hand side by $\Delta F$, as shown in Table 4:

$$\sum_i \frac{W_i^H \Delta F_i}{\Delta F} + \sum_i \frac{F_i^H \Delta W_i}{\Delta F} + \sum_i \frac{\Delta W_i \Delta F_i}{\Delta F} = 1. \tag{A4}$$

**Code and data availability**

HURDAT2 data can be downloaded from the NOAA's Hurricane Research Division (https://www.aoml.noaa.gov/hrd/hurdat/). ERA5 data can be obtained from the Copernicus C3S Date store (https://www.ecmwf.int/en/forecasts/datasets/reanalysis-datasets/era5). CMIP6 data can be obtained from the Earth System Grid Federation (https://esgf.llnl.gov/). Potential Intensity is calculated using the tcPyPI Python package (Gilford, 2021), available at https://github.com/dgilford/tcpyPI. TRACK can be downloaded from https://gitlab.act.reading.ac.uk/track/track, and version 1.5.2 is used for this study.

**Author contributions**

ES designed the study with input from RS, KH, AB, LS, and KB. ES performed TC identification and the analysis on cyclone tracks and environmental fields. All authors provided valuable feedback and shaped the study. KH performed cyclone tracking on ERA5 and the CMIP6 models. ES prepared the manuscript with input from all co-authors. RS, KH, AB, and LS obtained the funding for this project.

**Acknowledgements**

E. Sainsbury was funded by the Natural Environment Research Council (NERC) via the SCENARIO Doctoral Training Partnership (Grant NE/S0077261/1) with additional CASE funding from BP. R.S., K.H., A.B., and L.S. are supported by the U. K. National Centre for Atmospheric Science (NCAS) at the University of Reading. A.B. acknowledges funding from the PRIMAVERA project received from the European Commission (Grant 641727 of the Horizon 2020 research program), and



NERC funding through the North Atlantic Climate System Integrated Study (ACSIS) grant (NE/N018044/1). We thank Olivier Boucher and Thibaut Lurton (IPSL) for re-running the IPSL-CM6A-LR model provide us the SSP5-85 scenario data needed to include the model in this study. The IPSL-CM6 experiments were performed using the HPC resources of TGCC under the allocation of 2021-A0100107732 (project gencmip6), provided by GENCI (Grand Equipment National de Calcul Intensif). This work benefited from French state aid, managed by the ANR under the "Investissements d'avenir" programme (reference ANR-11-IDEX-0004-17-EURE-0006). S.B. is supported by public funding from the CEA and the EUR IPSL-Climate. L.S. and R.S. acknowledge funding from the NERC CANARI project (NE/W004984/1).

**Declaration**

The authors declare that they have no conflicts of interest.

600

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
