# Peer review of "Can low-resolution CMIP6 ScenarioMIP models provide insight into future European Post-Tropical Cyclone risk?"

_Weather and Climate Dynamics, 2022_

## Author Response (AR1)

**Response to Reviewer comments**

We thank the Editor for their time in finding reviewers for this manuscript, and thank the reviewers for their comments and suggestions which have improved the quality and clarity of the manuscript. All comments have been incorporated and are detailed below.

**Referee #1**

In this article the authors investigate the change in PTCs affecting Europe due to global warming in coarse resolution CMIP6 models. Because TC are intense small-scale systems the analyses of TCs and PTCs is usually done with the highest resolution models available. The authors of this article take the courage step to analyse TCs and the subsequent PTCs in present coarse CMIP6 ScenarioMIP runs.

Although courageous this choice is defendable because apart from being small scale and intense, TCs are characterized by their different structure such as warm core and can be identified in low resolution models although with smaller intensity and larger structure as already shown by Haarsma et al 1993.

The authors provide a thorough analysis of the change in PTCs affecting Europe and the different factors that play a role in this change, like change changes in TC genesis, recurvature and reintensification. For the first time the authors provide an analysis of the complete PTC life cycle and the changes due to global warming in climate models. This analyses is compared with HURDAT2 observations and ERA5 reanalyses highlighting that the CMIP6 ScenarioMIP models are able to capture the main processes that govern the evolution of TCs to PTCs. However, from these analyses it is also clear that the climate models still show large biases, which are not only due to the low resolution.

The strong point of this article is that it separates the different phases in the evolution of PTCs affecting Europe starting with the genesis of TCs in the Atlantic and investigates the different processes that play a dominant role in those phases and how these are affected by global warming, which has not been done before. I consider therefore this as an important paper, also for future studies with larger ensembles and higher resolution, for which this analysis can be repeated or serve as a guideline.

The scientific question how global warming will affect future European PTC risk is far from being answered. However, this study disentangles this question into other underlying relevant questions and provides useful analyses and preliminary answers.

The article is well written, and the analyses are done thoroughly. I recommend publication. I have only a few minor comments that I will discuss below.

1) The authors select only five models from the large CMIP6 set, based on a criterium outlined in the supplemental material. Five models is a rather small number for any statistical robust conclusions. The selection criterion is based on a TC frequency being equal or larger than observed hurricane frequency. No motivation for this selection criterium is given. I am wondering if a weaker criterium providing a larger ensemble would give similar

results and give more statistical robustness. Please provide arguments for this criterium choice.

In this manuscript, we chose to focus on models that reasonably represent the North Atlantic TC climatology as seen in observations and reanalyses. In best track data, we observe on average 12 TCs and 6.4 hurricanes per year in the North Atlantic. As climate models underestimate the intensity of TCs, it would be unreasonable to expect CMIP6 models to routinely produce hurricane-force TCs. As a result, we opted to retain CMIP6 models which have at least 6.4 TCs of any intensity in the North Atlantic.

A second consideration for this threshold is Europe-impacting PTC sample size. We choose to focus our analysis on a 30-year period at the end of the historical and future simulations to ensure we are able to detect any simulated trends. In reanalyses, we find that ~2 PTCs reach Europe per year (Baker et al. 2021), which is approximately 1/6th of all North Atlantic TCs. The threshold on TC activity was therefore also chosen to ensure that in each model, we had a large enough sample of Europe-impacting PTCs to perform meaningful statistical analysis.

We have repeated the analysis presented in Table 2 of the main manuscript on 10 additional CMIP6 models (the models which are shown using blue shading in in Figure S1), for which we have the future simulation data necessary for cyclone tracking and TC identification. While these models do not simulate a TC frequency comparable to observations, we agree that it is important to understand where the selected 5 CMIP6 models fall within the larger ensemble of CMIP6 models.

The statistics shown in Table 2 of the main manuscript are presented below (Tables 1 and 2) for the additional 10 CMIP6 models. As with the selected 5 models, we consider the same emission scenario, tracking algorithm, TC identification method and time periods.

This additional analysis is largely consistent with the results presented in the main manuscript. 9 of the 10 additional models show a significant (95% level) decrease in North Atlantic TC frequency of approximately 25-75%, which is offset to some degree by a projected increase in the likelihood of recurvature (positive change in all 10 models, significant in 4). As a result, 7 of the 10 models project an increase in the proportion of North Atlantic TCs reaching Europe, significant in 4 models (Table 2 in this document). 8 of 10 of the additional models project a decrease in Europe-impacting PTC frequency, significant in 4 models.

Interestingly, the EC-Earth3 stands out from the other models. It projects a significant increase in North Atlantic TC frequency, and more than a doubling in Europe-impacting PTC frequency. While the model formulations are different, EC-Earth also projected a significant increase in the frequency of hurricane-force PTCs over Europe in Haarsma et al. (2013).

The text in section 2.1 has been updated to provide some additional justification for the TC threshold (lines 98-100):

*"This threshold is chosen to ensure we focus on CMIP6 models which simulate North Atlantic TC frequency reasonably compared to observations, and to ensure the selected models have a sufficient sample size of TCs and Europe-impacting PTCs for meaningful statistical analysis"*

Tables 1 and 2 have been added to the supplementary material and referenced in the manuscript (lines 378-381):

*"The analysis of Table 2 is also repeated for 10 additional CMIP6 ScenarioMIP models which did not meet our selection criteria but have sufficient data available for cyclone tracking and TC identification (Tables S4, S5). These additional models support the results in Table 2 and highlight the robustness of the projected changes in North Atlantic TC frequency, likelihood of recurvature, and proportion of TCs reaching Europe in CMIP6 simulations. "*

| | $N_{TC}$ | | | $F_{rec}$ | | | $F_{Europe|rec}$ | | | $N_{Europe}$ | | |
|---|---|---|---|---|---|---|---|---|---|---|---|---|
| | Hist | SSP | Diff | Hist | SSP | Diff | Hist | SSP | Diff | Hist | SSP | Diff |
| ACCESS-CM2 | 104 | 49 | **-53%** | 0.41 | 0.71 | **73%** | 0.35 | 0.43 | 23% | 15 | 15 | 0% |
| BCC-CSM2-MR | 109 | 60 | **-45%** | 0.53 | 0.57 | 6% | 0.34 | 0.29 | -15% | 20 | 10 | **-50%** |
| EC-Earth3 | 45 | 65 | **47%** | 0.44 | 0.62 | **41%** | 0.32 | 0.37 | 14% | 6.3 | 14.9 | **133%** |
| MIROC-ES2L | 16 | 4 | **-73%** | 0.31 | 0.75 | 139% | 0.31 | 0 | -100% | 1.5 | 0 | -100% |
| MPI-ESM1-2-HR | 61 | 33 | **-47%** | 0.25 | 0.34 | 33% | 0.30 | 0.32 | 7% | 4.6 | 3.5 | -25% |
| MPI-ESM1-2-LR | 48 | 27 | **-44%** | 0.33 | 0.37 | 13% | 0.34 | 0.34 | -1% | 5.4 | 3.4 | **-40%** |
| MRI-ESM2-0 | 153 | 45 | **-70%** | 0.49 | 0.73 | **51%** | 0.30 | 0.39 | 30% | 22.4 | 13 | **-41%** |
| NESM3 | 146 | 74 | **-49%** | 0.46 | 0.69 | **50%** | 0.28 | 0.35 | 25% | 19 | 18 | -5% |
| NorESM2-LM | 77 | 58 | **-25%** | 0.52 | 0.59 | 14% | 0.40 | 0.44 | 9% | 16 | 15 | -6% |
| NorESM2-MM | 169 | 73 | **-57%** | 0.49 | 0.52 | 7% | 0.41 | 0.32 | -24% | 34 | 12 | **-64%** |

**Table 1**. As in Table 2 of the main manuscript, but for 10 additional (and poorer-performing with respect to North Atlantic TC frequency) CMIP6 ScenarioMIP models.

| | Proportion of TCs reaching Europe ( = $F_{rec}$ $F_{Europe|rec}$ ) | | |
|---|---|---|---|
| | Hist | SSP | Diff |
| ACCESS-CM2 | 0.14 | 0.31 | **122%** |
| BCC-CSM2-MR | 0.18 | 0.17 | -9% |
| EC-Earth3 | 0.14 | 0.23 | **61%** |
| MIROC-ES2L | 0.10 | 0.0 | -100% |
| MPI-ESM1-2-HR | 0.08 | 0.11 | 42% |

| | | | |
|---|---|---|---|
| MPI-ESM1-2-LR | 0.11 | 0.13 | 12% |
| MRI-ESM2-0 | 0.15 | 0.29 | **97%** |
| NESM3 | 0.13 | 0.24 | **87%** |
| NorESM2-LM | 0.21 | 0.26 | 24% |
| NorESM2-MM | 0.20 | 0.16 | -18% |

**Table 2**. Proportion of North Atlantic TCs reaching Europe in the historical (1984-2014) and future (2069-99) period under the SSP5-85 scenario (columns 2 and 3 respectively). Fractional change is shown in column 4. Bolded differences represent statistical significance at the 95% level using a bootstrap resampling method.

2)    Two of the models that are analysed (CNRM and HadGEM) have resolutions comparable to the HighResMIP resolution (~50 km atmosphere, 0.25° ocean). These two models are also analysed in Baker et al. 2022. So, the sentence at line 74-76 is not correct. HighResMIP models had a somewhat different protocol than ScenarioMIP (i.e. aersol forcing, land surface scheme). Also period and spin-up are differently. I assume that CNRM and HadGEM are based on the HighResMIP models and modified and used for the ScenarioMIP. Please explain.

While there are HighResMIP simulations available for CNRM and HadGEM, these are not utilised in this study. The simulations from CNRM and HadGEM used in this manuscript use the same science formulation as the other CMIP6 ScenarioMIP models which are used.

The text has been updated to clarify this (lines 103-107):

*"CNRM and HadGEM have a higher horizontal resolution in the atmosphere and ocean than the other selected models (Table 1). While CNRM and HadGEM also have HighResMIP simulations available, they use a different experimental protocol (e.g., different aerosol forcing and land surface scheme) and only run out to the year 2050. In this study we focus on the ScenarioMIP simulations for consistency with the other selected models."*

3)    Line 152-153 something went wrong in explaining the region of WEST. Later the regions are shown for instance in Fig. 7. May be mention that the figures are shown in the forthcoming plots?

The co-ordinates listed on lines 152-153 refer to the vertices of the WEST region. The text has been updated to make this clearer, and to also mention that the boundaries for the regions are shown in Figure 7.

4)    A notable difference in Fig.1 is the genesis over west Africa simulated by ERA5 and the

models, but not seen in HURDAT2. Is this spurious or are no observations available over Africa for HURDAT2? Please explain or discuss this.

TC identification occurs over the ocean after all cyclones are tracked, allowing us to identify the completely lifecycle associated with TCs, including the pre- and post-TC stages. Genesis is seen in ERA5 and CMIP6 models over west Africa because the tracking algorithm identifies the vorticity anomalies as soon as they form and before they are identified as TCs (TC precursors). For ERA5 and CMIP6 models, Figure 1 is therefore actually showing the locations in which the precursors to the TCs form and are identified (such as African Easterly Waves). In comparison, HURDAT2 genesis densities only show the locations of where TCs are first detected as TCs.

The text in the manuscript will be updated to reflect this change (lines 200-204):

*"Comparisons between HURDAT2 and ERA5/CMIP6 models should be made cautiously due to differences in how TCs are identified. For example, the cyclone detection and tracking scheme used in this study allows for the identification of TC precursors. Therefore, the genesis densities shown for the CMIP6 models and ERA5 represent the genesis density of the precursors to TCs, whereas the HURDAT2 genesis density shows where these precursors developed into TCs. This explains the differences in genesis density between CMIP6/ERA5 and HURDAT2 over west Africa."*

5) Line 232-233. To me this is no surprise as coarse resolution models tends to increase the size of TCs and reduce the pressure gradient. May be include this argument here?

Thank you for this comment. The text has been updated to include this argument.

6) Line 285. Maybe you can mention the numbers of Sainsbury et al 2020 here?

The text has been updated to include the numbers from Sainsbury et al. (2020), which are ~0.5% in the lowest intensity bin, and ~9% in the highest intensity bin.

7) Line 463-464. Here and in the supplemental material section 7 the effect of ensemble size is investigated. However, the ensemble spread is not discussed. That the ensemble size does not affect the basic results is reassuring, but I assume that there is considerable spread in TCs and also how much of those TCs recurve and translate into PTCs between individual ensemble members.

Table 3 (shown below) contains cyclone statistics for the two models which have more than one ensemble member: MIROC6 and IPSL-CM6A-LR. Brackets show the minimum and maximum for each of the statistics from the model's ensemble for the historical and future periods. For MIROC6, there are 10 historical members and 3 future members, and for IPSL-CM6A-LR there are 31 historical members, but only one future member.

While there is a large spread among ensemble members, the spread is considerably smaller than the projected changes in North Atlantic TC frequency ($N_{TC}$), the fraction of recurving TCs ($F_{Rec}$) and the fraction of North Atlantic TCs reaching Europe ($F_{Rec} F_{Eur|Rec}$) in these two models.

This table has been added to the supplementary material and referenced in section 3.2 (lines 376-378):

*"There is a large ensemble spread in the statistics presented in Table 2 in the MIROC and IPSL. However, the projected changes in North Atlantic TC frequency, likelihood of recurvature, and the proportion of North Atlantic TCs reaching Europe are considerably larger than the ensemble spread (Table S3)."*

|  | MIROC (10, 3) | | IPSL (31, 1) | |
| --- | --- | --- | --- | --- |
|  | Hist | SSP | Hist | SSP |
| $N_{TC}$ | (7.26, 8.29) | (5.23, 6.0) | (7.42, 8.84) | 3.16 |
| $F_{rec}$ | (0.37, 0.49) | (0.66, 0.71) | (0.16, 0.23) | 0.42 |
| $F_{Europe|rec}$ | (0.23, 0.36) | (0.31, 0.32) | (0.18, 0.44) | 0.32 |
| $N_{Europe}$ | (0.74, 1.29) | (1.06, 1.23) | (0.26, 0.65) | 0.42 |
| $F_{rec} F_{Europe|rec}$ | (0.09, 0.16) | (0.20, 0.23) | (0.03, 0.08) | 0.13 |

**Table 3**. Values of the ensemble minimum and maximum (min, max) North Atlantic TC frequency (row 2), fraction of recurving TCs (row 3), fraction of recurving TCs reaching Europe (row 4), Europe-impacting PTC frequency (row 5) and fraction of North Atlantic TCs reaching Europe (row 6) for the MIROC (columns 2, 3) and IPSL (columns 4, 5) in the historical and future simulations. The number of available ensemble members in the MIROC and IPSL are given in brackets in row 1.

An interesting result of this study is the important role of vertical wind shear for the development of PTCs. The important role of wind shear associated with El-Nino's for genesis and development of TCs is well known. This study also shows the large biases in wind shear in CMIP6 models. To this I want to add that recent studies also show that there is a mismatch during recent decades between the observed evolution of El-Nino and simulated by CMIP6 models (Seager et al. 2019). This will have an impact on vertical windshear over the North Atlantic and the genesis TCs and as a consequence also the PCT risk for Europe.

Thank you for raising this suggestion. The mismatch between observed and simulated ENSO in recent decades and the implications for future PTC risk to Europe has been included in the discussion, along with the suggested reference (lines 612-615):

*"Additionally, there is a mismatch between climate model projections and observations of the zonal temperature gradient in the tropical Pacific, which has implications for North*

*Atlantic vertical wind shear (Seager et al., 2019), which is important for TC genesis and may be important for the projected change in the likelihood of recurvature of North Atlantic TCs"*

Haarsma, R. J., Mitchell, J. F., & Senior, C. A. (1993). Tropical disturbances in a GCM. Climate Dynamics, 8(5), 247-257.

Seager, R., M. Cane, N. Henderson, D. E. Lee, R. Abernathey and H. Zhang (2019). 'Strengthening tropical Pacific zonal sea surface temperature gradient consistent with rising greenhouse gases'. In: Nature Climate Change 9.7, pp. 517–522. doi: 10.1038/s41558-019-0505-x.

**Referee #2 – Michiel Baatsen**

The authors present a study that looks at both the performance of CMIP6 models to represent tropical cyclones in the North Atlantic, and a future projection of recurving PTCs impacting Europe.

I think bost aspects of the study are relevant and new, as well as show useful results with opportunities for future work.

The methods are overall well explained, the manuscript is well-written and structurally sound, and the results properly assessed.

There is room for some improvement in the motivation of model choice and general assessment of their TC skill, to better understand their present and future skill regarding North Altantic TCs.

Some more detailed comparison to existing studies using high resolution models in particular would be useful, showing which of the expected trends are consistent and therefore useful between different sets of models.

This may be at least partly the subject of some previous or future work, but in that case it should be better clarified.

Regardless, the manuscript is of high quality and would only require limited adjustments for publication.

**General comments:**

- A selection of CMIP6 models is used, based on the present TC frequency over the North Atlantic Ocean. While this is probably a good measure, it is tough to say whether these models accurately show TC frequency for the right reasons and even more so if they are able to correctly show future trends.

  Especially with contrasting mechanisms determining the trend, this may be an important concern. It would therefore be useful to show some general metrics of these models, e.g. how they reproduce overal TC formation/intensification measures, or how they compare versus (limited) available high resolution simulations.

Additional analysis has been undertaken to investigate the genesis density in 10 additional CMIP6 models, to try and understand why the selected models are better able to capture TC frequency. This is shown below (Figure 1) in response to specific comment 3.

Additionally, section 3.1.3 has been updated to provide a more detailed comparison between the selected CMIP6 models and results from available high-resolution simulations (Baker et al. 2022; Vidale et al. 2021; Roberts et al. 2020), with a particular focus on the large-scale environment (lines 293-306):

*"Despite clear model biases, the selected CMIP6 models represent many features of the observed TC climatology, with spatial patterns and frequencies in qualitative agreement with observations. TC frequency, seasonal cycle, and spatial distribution in these selected CMIP6 models are comparable to those found in higher-resolution modelling studies, such as Climate-SPHINX (Vidale et al., 2021), UPSCALE (Roberts et al., 2015) and HighResMIP-PRIMAVERA (Roberts et al., 2020a; Haarsma et al., 2016; Baker et al., 2022), which used the same tracking and identification scheme. However, many high-resolution climate models are able to simulate TCs with intensities greater than 50ms$^{-1}$ (Baker et al., 2022; Vidale et al., 2021), unlike all selected CMIP6 models except CNRM in this study. Many high-resolution models contain biases in their large-scale environment, but in most cases these biases are not consistent between models (Roberts et al., 2020a). This is also true for the selected CMIP6 models with the exception of vertical wind shear, which is too high in the MDR in all selected models except CNRM (Fig. S4). Improving model resolution does not systematically improve historical biases in the large-scale environmental fields correlated with TC genesis and intensification, but does reduce historical biases in TC frequency and improves the spatial distribution of TCs, particularly in the MDR (Roberts et al., 2020a; Vidale et al., 2021; Baker et al., 2022). It is therefore possible that the lack of genesis in the western MDR in many of the selected CMIP6 models is the result of too much vertical wind shear and insufficient model resolution. "*

- I would like to see some more information regarding the thresholds used to define a TC, both for tracking as for impacting Europe (does this include wind speed, or just vorticity?).

  Although previous methods are mostly being used and well documented, this could help avoid some confusion going through the results.

There are two steps involved in the process:

Cyclone tracking (all cyclones) is performed first. In this step, relative vorticity fields (850hPa for CMIP6, the 600-850hPa vertical average for ERA5) are spectrally filtered to T42 (CMIP6) or T63 (ERA5) and planetary waves (wavenumber < 6) are removed. All vorticity features exceeding the threshold $0.5 \times 10^{-5}$ s$^{-1}$ are initialised into cyclone tracks using a nearest neighbour method. The tracks are then refined by minimising a cost function for track smoothness (Hodges 1995). No wind speed thresholds are used, and the vorticity threshold is deliberately low to ensure all possible features are tracked. Cyclones lasting for longer than two days are retained.

The second step is TC identification, which is performed on the tracks identified by cyclone tracking (step 1). For a cyclone track to be identified as a TC the following criteria must be met:

1) The first point in the cyclone track (genesis) must be equatorward of 30N
2) The T63 relative vorticity must exceed $6 \times 10^{-5}$ s$^{-1}$
3) The difference in T63 relative vorticity between the 850hPa and 200hPa levels must exceed $6 \times 10^{-5}$ s$^{-1}$ to indicate the existence of a warm core via thermal wind balance

4) A T63 relative vorticity signature must exist at each pressure level between 250 and 850hPa to indicate a coherent vertical structure throughout the troposphere

Criteria 2-4 must be met for at least 4 consecutive time steps (one day) over the ocean. In ERA5, there are many available vertical levels for relative vorticity. In this case, criteria 4 is applied to the 850-, 700-, 600-, 500-, 400, 300- and 250 hPa pressure levels. Less vertical levels are available for CMIP6 models, and so the criteria are only applied to the 850-, 500- and 250hPa levels. While cyclone tracking is performed at T42 and T63 for CMIP6 and ERA5 respectively, TC identification is performed at T63 in all cases, but with less vertical levels when applied to CMIP6 models.

No wind speed thresholds are utilised during cyclone tracking or TC identification. The only additional requirement placed on cyclones reaching Europe is the geographical region. TC identification methods which use wind speed thresholds often need to modify the threshold according to model resolution (Walsh et al., 2015), whereas the methodology used here aims to be as resolution-independent as possible.

The methods section of the manuscript has been updated to include this additional information on the thresholds used, and to highlight that tracking and TC identification are performed just using relative vorticity data (lines 139-156).

- In the results section, the first paragraph of most subsections is more of a motivation/methods part. This can be a deliberate choice, but makes the results section slightly more tedious and less focussed.

This manuscript contains many different metrics which can be somewhat challenging to follow. The additional motivation at the start of many subsections was a deliberate choice by the authors to ensure that justification was given for the upcoming analysis. However, we appreciate that this does make the manuscript longer and less focussed, and so these paragraphs have been revisited and a number of sentences have been removed.

- The results cover different statistics about TC recurvature, frequency and track, which is very useful. There is, however, an important distinction between considering a fraction of North Atlantic TCs (recurving, impacting Europe etc. e.g. Table 2, Figure 10), or rather cyclones impacting Europe as a start (i.e. how many of those are PTCs e.g. Figures 5,9). This distinction could be made clearer and considered more deliberately at times, to avoid confusion.

The methods (section 2.4) have been updated to clearly signpost that the metrics presented in this manuscript are conditioned in two different ways; cyclone statistics conditional on impacting Europe for Figures 5 and 9, and cyclone statistics conditional on being TCs for the remainder of the analysis (lines 179-183):

*"It should be noted that metrics are conditioned in two different ways throughout the results section. For example, in Figures 5 and 9 metrics are presented which are conditional on cyclones reaching Europe (e.g., given that a set of cyclones reach Europe, what fraction are PTCs?), whereas in the rest of the manuscript metrics are conditioned on cyclones being TCs (e.g., given that a set of cyclones are TCs, what fraction reach Europe as PTCs?). "*

**Specific comments:**

- L46 and following: it is important to clarify that the Haarsma and Baatsen studies only focus on PTCs impacting Europe with storm-force winds, which is only a minor fraction of all PTCs.

The introduction has been updated to make this clarification.

- L123 & L130: Besides earlier work and data limitations, could you motivate the different choices between the datasets and the potential limitations for the results? (e.g. weaker vorticity signatures at T42)

The main aim of section 3.1. was to identify how well the selected CMIP6 models capture key features of the North Atlantic TC and PTC climatologies. HURDAT2 data can be used for this to an extent, but due to data inhomogeneities (particularly post-ET) we felt that it was also sensible to compare the CMIP6 models to a reanalysis.

We had two choices in this respect:

1) Perform the cyclone tracking and TC identification on ERA5 using the exact same data levels and methodology as CMIP6 (i.e., T42 filtering, TC identification using vorticity on fewer pressure levels).

2) Perform the cyclone tracking and TC identification on ERA5 using the 'state-of-the-art' method – using all of the available data, leading to TC climatology which is likely closer to the 'truth', but is less directly comparable to CMIP6 models.

Ultimately the goal of section 3.1 was to identify how well the CMIP6 models simulate TC and PTC statistics compared to observations, and the decision was therefore made to perform the tracking and TC identification in ERA5 using all possible data. This difference in methodology and data has been tested previously by other scientists (in unpublished work), and differences were found to be small (not shown).

The main limitations are a weaker vorticity signature at T42 as you mention, but also a less coherent tracking. This is because there is no vertical averaging of the vorticity fields before the filtering to T42 resolution. This mainly affects the pre-TC stage of the cyclone's lifecycle.

The text in the methods section has been updated to mention the lack of sensitivity that the method has to cyclone statistics (lines 154-156).

- L166-184: It would be useful to perform some of these analyses (or just e.g. the GPI) on the full CMIP6 ensemble, to get a better understanding why the selected models are better representing TC frequency.

Genesis density has been computed for 10 additional CMIP6 models for which we have sufficient data in the historical and future periods (blue shades boxes, Figure S1) and are shown below in Figure 1. The same time period (1984-2014), cyclone tracking, and TC identification methods are used as in the main manuscript for consistency. The main bias

appears to be in the MDR, where 8 of the 10 additional models have low – if any - TC activity.

It is not clear why the selected models represent MDR TC activity better than many of the additional models, but horizontal resolution, the large-scale environment, and TC seeds (frequency, intensity, and conversion rate) may all play a role. Existing studies have looked in great detail at the simulation of the large-scale tropical environment (Han et al. 2022) in a larger ensemble of CMIP6 simulations, and additional work (in prep) is underway at other institutions to investigate this further. An investigation into why the selected CMIP6 models better capture TC activity than others may detract from the main goal of this paper (Europe-impacting PTCs) and is outside of the intended scope of this paper.

[Figure]

**Figure 1**. Genesis density for the historical (1984-2014) period in the 10 additional CMIP6 models shown in Figure S1 (blue shading) for which sufficient data is available for cyclone detection and tracking.

Section 3.1. has been updated to provide some more information (lines 211-216):

*"Additional analysis on 10 further CMIP6 models which did not meet our TC criteria (blue shaded boxes, Figure S1) showed very little (or no) genesis in the MDR, which is likely responsible for the low TC counts in these additional models (not shown). The cause(s) of the bias in TC activity in the MDR in these additional models is outside of the scope of this paper, but may be associated with insufficient horizontal resolution, differences in the representation of the large-scale circulation and tropical environment into which TCs are forming, and differences in the representation of TCs seeds (frequency, intensity and conversion rate)."*

- L208: To test whether the models dissipate existing TCs too rapidly or rather convert too few seeds into TCs, you can also consider the average track length and duration of the considered TCs.

Thank you for the suggestion. On average, MDR-forming TCs have a track length of 9700km and an average duration of 16.5 days in ERA5. In IPSL, this is just to 4200km, and 8.7 days. This information has been added to the text in section 3.1

- L219: Can you explain this underestimation, while models seem to represent overall TC frequency well? Are conditions too hostile in the north or rather the models unable to represent physical processes correctly towards ET?

Especially for cases of re-intensification after ET, it could be useful to check SST fields (and gradients) in the models.

Europe-impacting PTC frequency is underestimated because TCs forming in the MDR are unlikely to recurve in the models compared to ERA5. Therefore, there are less TCs reaching a region of baroclinicity that may facilitate ET and future reintensification. This is highlighted in column 3 of Table 3 of the manuscript, which shows that in all models except the CNRM, the fraction of recurving TCs is too low. This may be associated with the positive MDR wind shear bias in all selected CMIP6 models except CNRM, which may lead to MDR TCs which are too weak and short-lived.

SSTs (and gradients) in the historical period in the selected models were plotted and compared to ERA5 but were not helpful in explaining the low bias in Europe-impacting PTC frequency compared to ERA5. However, the plot is included below for completeness. The model with the greatest SST gradients near the entrance to the midlatitude storm track (KIOST) is also the model with the greatest proportion of recurving TCs reaching Europe (Fig. 3), possibly highlighting a more favourable environment for reintensification post-ET.

The text has been updated to explain the low-bias in Europe-impacting PTC frequency in the selected models, and to mention the additional insights provided by the analysis of SST fields (and gradients).

[Figure]

**Figure 2**. Hurricane season sea surface temperatures in the historical (1984-2014) period in the five selected CMIP6 models (a-e) and ERA5 (f). The number of ensemble members used is shown to the top-right of each panel.

[Figure]

**Figure 3**. SSTs as a function of latitude at a number of different longitude boundaries in selected CMIP6 models and ERA5 (top row). Bottom row shows relative SSTs as a function of latitude, where relative SSTs are defined as the SST relative to the SST at 35N (in each given model, at the given longitude boundary).

- L237: It would surprise me if baroclinic forcing would play a major role over the Gulf of Mexico during much of the Hurricane season, as would model resolution differences between 10-20N and 25-30N. What about e.g. biases in SST, RH or wind shear?

Figure S4 shows that the IPSL has a positive wind shear bias in the MDR and a slightly negative potential intensity bias (both relative to ERA5), but these biases are not larger than found in the other four selected CMIP6 models. Therefore, it is unlikely that these biases alone explain the difference in TC representation in the IPSL compared to the other selected models.

Many TCs forming along the US East Coast are baroclinically influenced, unlike those forming in the MDR (Fig 1, Elsner et al. 1996). Similarly, effective resolution is greater at higher latitudes. While the hypotheses presented on line 237 are rather speculative, they may have some association with the deficiencies of MDR TCs in the IPSL, however a dedicated study would likely be needed to quantify their importance.

The text in this paragraph has been updated to make it clear that these hypotheses are speculative and that other factors may also play a role. The text mentioning that baroclinicity may be important in the Gulf of Mexico has been removed. The text has also been updated to refer to Figure S4 to highlight that environmental biases in IPSL are no larger than the other selected models.

- L256: As mentioned in the general comments, I believe a more detailed comparison would be insightful here, especially regarding the environmental conditions for TC formation and maintenance.

This paragraph has been updated to provide a more detailed comparison of cyclone statistics and environmental conditions in the selected CMIP6 models and high-resolution studies, and this text has been quoted in response to general comment #1.

- L324: This seems to happen mostly towards the end of the 21st century, following an increase in intense storms that originate from the tropics but are not PTCs (Baatsen et al 2015).

Thank you for the clarification. This sentence has been removed.

**Figures:**

- Figure 1: The HURDAT2 contours are rather tough to see, particularly as they conflict with the coastlines. Consider e.g. making the coastlines a lighter gray for clarity.

Coastlines in Figures 1 and 2 have been re-plotted in gray.

- Figure 1&2: As the lower values are masked out, it would make more sense to use a reversed colouring i.e. darker colours for higher values?

  The highest values run beyond the scale, which is probably chosen to make for an easy overview. Consider adding some larger steps to the top of the scale to be able to distinguish the maxima as well. Right now, one cannot tell whether this value is 10 or more like 50.

Colour scale for Figures 1 and 2 (and 7) have been reversed, and the range of the colour bars have been extended.

- Figure 5: It would be useful to include some information regarding total counts as well, both on the cyclone frequencies as on wind speed percentiles.

  It is unclear to me how the percentage given in each panel relates to the shown bins: I understand this is the total fraction of Europe-impacting cyclones that are PTCs during Hurricane season. In that case, the number should correspond to that of the 0th-percentile, as this would include all storms. Clearly it does not, so I am missing something here.

The total number of Europe-impacting PTCs and MLCs during hurricane season have been added to each panel. The plot looks quite messy when adding the counts for each percentile-based bin, and so these have been added to a table in the supplementary material.

The bars in Figure 5 represent the proportion of hurricane-season-forming, Europe-impacting cyclones in a given intensity bin (the lower bound of which is specified by the x label) which are PTCs. The $0^{th}$ percentile therefore would correspond to the shown percentages (but none of the bars represent this). The first set of bars in each panel are not considering the $0^{th}$ percentile, but all storms in the intensity bin ranging from the $0^{th} - 20^{th}$ percentile.

The percentage shown in each panel can be calculated by taking the sum over all bins of the PTC and MLC counts, as there is no overlap between the bins. The text has been updated to make this clearer (lines 328-332):

*"To ensure that the sample size remains reasonable across bins and across models, we bin the cyclones based on percentiles of the combined distribution. For each model, we combine the (Europe-impacting) PTC and MLC cyclone tracks over both time periods (historical and future) and calculate percentiles of this joint distribution of their maximum 10m wind speeds over Europe. These percentiles are then used to bin the data. The bins therefore correspond to 0-20th, 20-40th, 40-60th, 60-80th, 80-90th, 90-95th, and > 95th percentiles."*

A typo in the percentage for Fig. 5c has also been corrected (1.93% -> 0.93%).

- Figure 6: This is an insightful and nice overview figure, it could benefit from a CMIP6 ensemble mean comparison.

  Figure S4 is a great addition in the supplement, so it would make sense referring here as well.

The ensemble mean has been added to Fig. 6, and Figures S4 referenced in the text.

- L405: Term 3 seems a bit unintuitive, being the combination of terms 1 and 2? While still referring to the appendix for more detailed information, it would be nice to have a bit more explanation on how this combination works.

Term 3 is rather unintuitive and represents the nonlinear combination of 1) a shift in TC genesis, and 2) a change in the likelihood of recurvature within the three regions. This term arises from the expansion of equation A2 when substituting $W_i^S$ with $W_i^H + \Delta W_i$ and $F_i^S$ with $F_i^H + \Delta F_i$. This term is usually small, with either term 1 or 2 dominant, but is included for completeness.

The text has been updated to provide this extra context on how term 3 arises.

**Technical remarks/typos:**

- L17: similar occurring twice

This has been fixed.

- L276: 'is show'

This has been fixed.

- L291: consider making the list numbers bold font for clarity

List numbers have been changed to bold.

- Table 3: this is quite an extensive table, is there a possibility to include some (simple) coloured shading to the cells to facilitate visual comparison?

WCD guidelines recommend avoiding the use of colour-shading of cells. While we agree that this would help readability, we have contacted the Editor for their opinion on this and have recommended avoiding the use of colour shading.

Baker, A., Roberts, M. J., Vidale, P. L., Hodges, K. I., Seddon, J., Vanniere, B., Haarsma, R. J., Schiemann, R. K. H., Kapetanakis, D., Tourigny, E., Lohmann, K., Roberts, C. D., and Terray, L.: Extratropical transition of tropical cyclones in a multiresolution ensemble of atmosphere-only and fully coupled global climate models, J. Clim., 35, 5283–5306, https://doi.org/10.1175/JCLI-D-21-0801.1, 2022.

Elsner, J. B., Lehmiller, G. S., and Kimberlain, T. B.: Objective Classification of Atlantic Hurricanes, J. Clim., 9, 2880–2889, https://doi.org/10.1175/1520-0442(1996)009<2880:OCOAH>2.0.CO;2, 1996.

Han, Y., Zhang, M. Z., Xu, Z., and Guo, W.: Assessing the performance of 33 CMIP6 models in simulating the large-scale environmental fields of tropical cyclones, Clim. Dyn., 58, 1683–1698, https://doi.org/10.1007/s00382-021-05986-4, 2022.

Hodges, K. I.: Feature Tracking on the Unit Sphere, Mon. Weather Rev., 123, 3458–3465, https://doi.org/10.1175/1520-0493(1995)123<3458:ftotus>2.0.co;2, 1995.

Roberts, M. J., Camp, J., Seddon, J., Vidale, P. L., Hodges, K., Vanniere, B., Mecking, J., Haarsma, R., Bellucci, A., Scoccimarro, E., Caron, L. P., Chauvin, F., Terray, L., Valcke, S., Moine, M. P., Putrasahan, D., Roberts, C., Senan, R., Zarzycki, C., and Ullrich, P.: Impact of model resolution on tropical cyclone simulation using the HighResMIP-PRIMAVERA multimodel ensemble, J. Clim., 33, 2557–2583, https://doi.org/10.1175/JCLI-D-19-0639.1, 2020.

Vidale, P. L., Hodges, K., Vannière, B., Davini, P., Roberts, M. J., Strommen, K., Weisheimer, A., Plesca, E., and Corti, S.: Impact of stochastic physics and model resolution on the simulation of Tropical Cyclones in climate GCMs, J. Clim., 1–85, https://doi.org/10.1175/jcli-d-20-0507.1, 2021.

Walsh, K. J. E., Camargo, S. J., Vecchi, G. A., Daloz, A. S., Elsner, J., Emanuel, K., Horn, M., Lim, Y. K., Roberts, M., Patricola, C., Scoccimarro, E., Sobel, A. H., Strazzo, S., Villarini, G., Wehner, M., Zhao, M., Kossin, J. P., La Row, T., Oouchi, K., Schubert, S., Wang, H., Bacmeister, J., Chang, P., Chauvin, F., Jablonowski, C., Kumar, A., Murakami, H., Ose, T., Reed, K. A., Saravanan, R., Yamada, Y., Zarzycki, C. M., Luigi Vidale, P., Jonas, J. A., and Henderson, N.: Hurricanes and climate: The U.S. Clivar working group on hurricanes, Bull. Am. Meteorol. Soc., 96, 997–1017, https://doi.org/10.1175/BAMS-D-13-00242.1, 2015.

---

## Author Response (AR2)

Can low-resolution CMIP6 ScenarioMIP models provide insight into future European Post-Tropical Cyclone risk?

**Technical corrections**

We would like to that Michiel Baatsen for their further comments to improve the readability of some figures. These changes have been implemented.

Reviewer comment:

The revised manuscript and the author's comments have resolved, addressed and/or clarified all of the potential issues raised during revision. In this sence, the article is ready for publication.
That said, some of the figures could use some adjustments for clarification. Especially figures 5, 6, 7 and 9 need some bigger fonts and at times thicker lines for readability. Consider rearranging some of the multi-panel figures (6,7 in particular, 1,2 potentially) to allow some larger panels and the opportunity to leave out repetitive colour scales. In terms of contents, scales and colour maps, the figures work very well as they are.

The panels comprising Figures 1, 2, 6 and 7 have been rearranged to allow for bigger panels and to reduce the number of colour bars that need to be shown. In addition, title font size, colour bar tick size and label font sizes have also been increased for figures 1, 2, 5, 6 and 7. The bar widths for Figure 5 have also been increased, and the text font size on the panels has been increased. The text in the figure captions has been updated to reflect these changes where necessary, and in-text references to specific panels of figures have been updated to reflect the new panel labels on figures where panels have been re-arranged.